# META-EVOLVE: CONTINUOUS ROBOT EVOLUTION FOR ONE-TO-MANY POLICY TRANSFER

**Xingyu Liu, Deepak Pathak, Ding Zhao**
Carnegie Mellon University
`{xingyul3,dpathak,dingzhao}@andrew.cmu.edu`

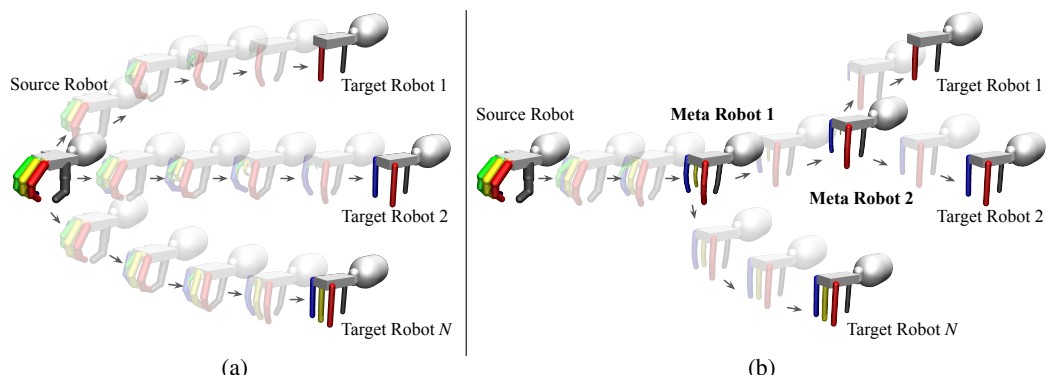

Figure 1: (a) **REvolveR and HERD** (Liu et al., 2022a;b) are methods for transferring policy between a pair of robots using continuous robot evolution. Therefore, to transfer a policy on the source robot to multiple target robots, they must launch multiple independent runs for each target robot. (b) **Our Meta-Evolve** uses continuous robot evolution to transfer an expert policy from the source robot to each target robot through an evolution tree defined by the connections of multiple "meta robots", i.e. tree-structured evolutionary robot sequences.

## ABSTRACT

We investigate the problem of transferring an expert policy from a source robot to multiple different robots. To solve this problem, we propose a method named *Meta-Evolve* that uses continuous robot evolution to efficiently transfer the policy to each target robot through a set of tree-structured evolutionary robot sequences. The robot evolution tree allows the robot evolution paths to be shared, so our approach can significantly outperform naive one-to-one policy transfer. We present a heuristic approach to determine an optimized robot evolution tree. Experiments have shown that our method is able to improve the efficiency of one-to-three transfer of manipulation policy by up to $3.2\times$ and one-to-six transfer of agile locomotion policy by $2.4\times$ in terms of simulation cost over the baseline of launching multiple independent one-to-one policy transfers. Supplementary videos available at the project website: `https://sites.google.com/view/meta-evolve`.

## 1 INTRODUCTION

The robotics industry has designed and developed a large number of commercial robots deployed in various applications. How to efficiently learn robotic skills on diverse robots in a scalable fashion? A popular solution is to train a policy for every new robot on every new task from scratch. This is not only inefficient in terms of sample efficiency but also impractical for complex robots due to a large exploration space. Inter-robot imitation by statistic matching methods that optimize to match the distribution of actions (Ross et al., 2011), transitioned states (Liu et al., 2019; Radosavovic et al., 2020), or reward (Ng et al., 2000; Ho & Ermon, 2016) could be possible solutions. However, they can only be applied to robots with similar dynamics to yield optimal performance.

Recent advances in evolution-based imitation learning (Liu et al., 2022a;b) inspire us to view this problem from the perspective of policy transferring from one robot to another. The core idea is to interpolate two different robots by producing a large number of intermediate robots between them which gradually evolve from the source robot toward the target robot. These continuously and gradually evolving robots act as the bridge for transferring the policy from the source to the

target robot. The source robot is usually selected as a robot such that it is easy to collect sufficient demonstrations to train a high-performance expert policy, e.g. a Shadow Hand robot that can be trained from large-scale human hand demonstration data (Grauman et al., 2022; Damen et al., 2018). While continuous robot evolution has shown success in learning challenging robot manipulation tasks (Liu et al., 2022a), the policy transfer is limited to being between a pair of robots. As illustrated in Figure 1(a), for $N$ different target robots where $N > 1$, it requires launching $N$ independent runs of robot-to-robot policy transfer and is not scalable. How can one efficiently transfer a well-trained policy from one source robot to multiple different target robots?

The history of biodiversity provides inspirations for this problem. In the biological world, similar creatures usually share the same ancestors in their evolution history before splitting their ways to form diverse species (Darwin, 1859). The same holds true for the robotic world. When robots are designed to complete certain tasks, they often share similar forms of morphology and dynamics to interact with other objects in similar ways. Examples include robot grippers that are all designed to close their fingers to grasp objects and multi-legged robots that are all designed to stretch their legs for agile locomotion. Therefore, to transfer the policy to $N$ different target robots, it may be possible to find common robot "ancestors" and share some parts of the robot evolution paths among the target robots before splitting their ways to each target robot. In this way, the cost of exploration and training during policy transfer can be significantly reduced. The idea is illustrated in Figure 1(b).

We propose a method named *Meta-Evolve* to instantiate the above idea. Given the source and multiple target robots, our method first matches their kinematic tree topology. This allows the source robot and all target robots to be represented in the same high-dimensional continuous space of physical parameters and also allows generating new intermediate robots. To share the evolution paths among multiple target robots, it requires defining a set of robot evolution sequences that are organized in a tree structure with the source robot being the root node. We propose a heuristic approach to determine the robot evolution tree within the parameter space by minimizing the total cost of training and exploration during policy transfer. We formulate the problem as finding a Steiner tree (Steiner, 1881; Gilbert & Pollak, 1968) in the robot parameter space that interconnects the source and all target robots. Our algorithm then decides evolution path splitting based on the evolution Steiner tree.

We showcase our Meta-Evolve on three Hand Manipulation Suite manipulation tasks (Rajeswaran et al., 2018) where the source robot is a five-finger dexterous hand and the target robots are three robot grippers with two, three, and four fingers respectively. Our Meta-Evolve reduces the total number of simulation epochs by up to $3.2\times$ compared to pairwise robot policy transfer baselines (Liu et al., 2022a;b) to reach the same performance on the three target robots. When applied to one-to-six policy transfer on four-legged agile locomotion robots, our Meta-Evolve can improve the total simulation cost by $2.4\times$. This shows that our Meta-Evolve allows more scalable inter-robot imitation learning.

## 2 PRELIMINARY

**Notation** We use bold letters to denote vectors. Specially, $\mathbf{0}$ and $\mathbf{1}$ are the all-zero and all-one vectors with proper dimensions respectively. We use $\odot$ to denote the element-wise product between vectors. We use $\boldsymbol{\theta}$ with subscripts to denote robot *physical* parameters and $\boldsymbol{\alpha}$ and $\boldsymbol{\beta}$ with subscripts to denote *evolution* parameters. We use $\mathrm{MAX}(\cdot)$ and $\mathrm{MIN}(\cdot)$ to denote element-wise maximum and minimum of a set of vectors respectively. $||\cdot||_p$ denotes the $L^p$ vector norm. $|\cdot|$ denotes the set cardinality.

**MDP Preliminary** We consider a continuous control problem formulated as Markov Decision Process (MDP). It is defined by a tuple $(\mathcal{S}, \mathcal{A}, \mathcal{T}, R, \gamma)$, where $\mathcal{S} \subseteq \mathbb{R}^S$ is the state space, $\mathcal{A} \subseteq \mathbb{R}^A$ is the action space, $\mathcal{T} : \mathcal{S} \times \mathcal{A} \to \mathcal{S}$ is the transition function, $R : \mathcal{S} \times \mathcal{A} \to \mathbb{R}$ is the reward function, and $\gamma \in [0, 1]$ is the discount factor. A policy $\pi : \mathcal{S} \to \mathcal{A}$ maps a state to an action where $\pi(a|s)$ is the probability of choosing action $a$ at state $s$. Suppose $\mathcal{M}$ is the set of all MDPs and $\rho^{\pi, M} = \sum_{t=0}^{\infty} \gamma^t R(s_t, a_t)$ is the episode discounted reward with policy $\pi$ on MDP $M \in \mathcal{M}$. The optimal policy $\pi_M^*$ on MDP $M$ is the one that maximizes the expected value of $\rho^{\pi, M}$.

**REvolveR and HERD Preliminary** Liu et al. (2022b) proposed a technique named REvolveR for transferring policies from one robot to a different robot. Given a well-trained expert policy $\pi_{M_S}^*$ on a source robot $M_S \in \mathcal{M}$, its goal is to find the optimal policy $\pi_{M_T}^*$ on another target robot $M_T \in \mathcal{M}$. The core idea is to define a sequence of intermediate robots through linear interpolation of physical parameters and sequentially fine-tune the policy on each intermediate robot in the sequence. Liu et al.

(2022a) proposed HERD, which extends the idea to robots represented in high-dimensional parameter space, and proposes to optimize the robot evolution path together with the policy. Concretely, given source and target robots $M_S, M_T \in \mathcal{M}$ that are parameterized in $D$-dimensional space, HERD defines a continuous function $F : [0,1]^D \to \mathcal{M}$ where $F(\mathbf{0}) = M_S, F(\mathbf{1}) = M_T$. Given the physical parameters of the source and target robots $\boldsymbol{\theta}_S, \boldsymbol{\theta}_T \in \mathbb{R}^D$ respectively, for any evolution parameter $\boldsymbol{\alpha} \in [0,1]^D$, $F(\boldsymbol{\alpha})$ defines an intermediate robot whose physical parameters are $\boldsymbol{\theta} = (1-\boldsymbol{\alpha})\odot\boldsymbol{\theta}_S+\boldsymbol{\alpha}\odot\boldsymbol{\theta}_T$. Then an expert policy $\pi^*_{F(\mathbf{0})}$ on the source robot $F(\mathbf{0})$ is optimized by sequentially interacting with each intermediate robot in the sequence $F(\boldsymbol{\alpha}_1), F(\boldsymbol{\alpha}_2), \ldots, F(\boldsymbol{\alpha}_K)$ where $\boldsymbol{\alpha}_K = \mathbf{1}$, until the policy is able to act (near) optimally on each intermediate robot. At robot $F(\boldsymbol{\alpha}_k)$, the optimization objective for finding the next best intermediate robot $F(\boldsymbol{\alpha}_{k+1}) := F(\boldsymbol{\alpha}_k + \boldsymbol{l}_k)$ is

$$\max_{||\boldsymbol{l}_k||_2=\xi} \quad \max_{\pi} \quad \mathbb{E}[\rho^{\pi, F(\boldsymbol{\alpha_k}+\boldsymbol{l_k})}] - \lambda/2 \cdot ||\mathbf{1} - (\boldsymbol{\alpha_k} + \boldsymbol{l_k})||_2^2 \tag{1}$$

which optimizes both the expected reward $\mathbb{E}[\rho^{\pi, F(\boldsymbol{\alpha_k}+\boldsymbol{l_k})}]$ and the $L^2$ distance to the target robot $||\mathbf{1} - (\boldsymbol{\alpha_k} + \boldsymbol{l_k})||_2$. For all $k$, the evolution step size $\xi = ||\boldsymbol{l}_k||_2$ is small enough so that each policy fine-tuning step is a much easier task. The idea is illustrated in Figure 1(a).

## 3 ONE-TO-MANY ROBOT-TO-ROBOT POLICY TRANSFER

### 3.1 PROBLEM STATEMENT

We investigate a new problem of transferring an expert policy from one *source* robot to multiple *target* robots. Formally, we consider a source robot $M_S \in \mathcal{M}$ and $N$ target robots $M_{T,1}, M_{T,2}, \ldots, M_{T,N} \in \mathcal{M}$ respectively. We assume the state space $\mathcal{S}$, and action space $\mathcal{A}$, reward function $\mathcal{R}$ and discount factor $\gamma$ of $M_S$ and all $M_{T,i}$ are shared and the difference is their transition dynamics $\mathcal{T}$. Given a well-trained expert policy $\pi^*_{M_S}$ on a source robot $M_S$, the goal is to find the optimal policies $\pi^*_{M_{T,i}}$ on each of the target robot $M_{T,i}$. We would like to investigate using the information in $\pi^*_{M_S}$ to improve the learning of $\pi_{M_{T,i}}$ and study how the learning of each individual $\pi_{M_{T,i}}$ can help each other.

We approach this problem by defining multiple *meta* robots $M_{\text{Meta},l} \in \mathcal{M}$ that shares the same state and action space as $M_S$ and all $M_{T,i}$. The meta robots $M_{\text{Meta},l}$ are designed such that $M_{\text{Meta},l}$ interconnects $M_S$ and all $M_{T,i}$ in an efficient way. Therefore, instead of repeating the process of one-to-one policy transferring for $N$ times, we can transfer source robot policy $\pi_S$ by going through the interconnection formed by meta robots $M_{\text{Meta},l}$ to reach each individual $M_{T,i}$.

### 3.2 MULTI-ROBOT MORPHOLOGY MATCHING AND INTERMEDIATE ROBOT GENERATION

Our problem setting and our proposed solution are based on an assumption that even if the source robot $M_S$ and target robots $M_{T,i}$ are different in action and state spaces, they can still be mapped to the same state and action space based on which the intermediate robots can be defined. The assumption is true for a pair of robots as shown in Liu et al. (2022a) and Liu et al. (2022b) where the intermediate robots are produced by robot morphology matching and kinematic interpolation. In this subsection, we show that this assumption can be extended to more than two robots.

**Kinematic Tree Matching** The topology of the kinematic tree of a robot describes the connection of the bodies and joints and reflects the kinematic behavior of the robot. Given two different robots with different kinematic tree, it has been shown in Liu et al. (2022b) that their kinematic trees can be matched by adding extra nodes and edges. This step can be extended to $N$ robots when $N > 2$. As illustrated in Figure 2(a), by matching proper root and leaf nodes, the matched kinematic tree is essentially a graph union of the kinematic trees of all $N$ robots. This means each robot needs to create additional bodies, joints and motors, though they may be all zero in numbers at the beginning.

The above kinematic tree matching process can be automated by an algorithm that achieves the best matching of nodes and edges across $N$ robots. However, in practice, we would like the matching process to include reasonable human intervention with enough robotics knowledge, e.g. matching human hand fingers to robot gripper fingers such that the knuckle joints are matched correctly.

**Physical Parameter Interpolation** After kinematic tree matching, the state and action space of the robots are matched. The difference in the robot transition dynamics is now only due to the differences in physical parameters, such as shapes and mass of robot bodies, gain and armature of joint motors,

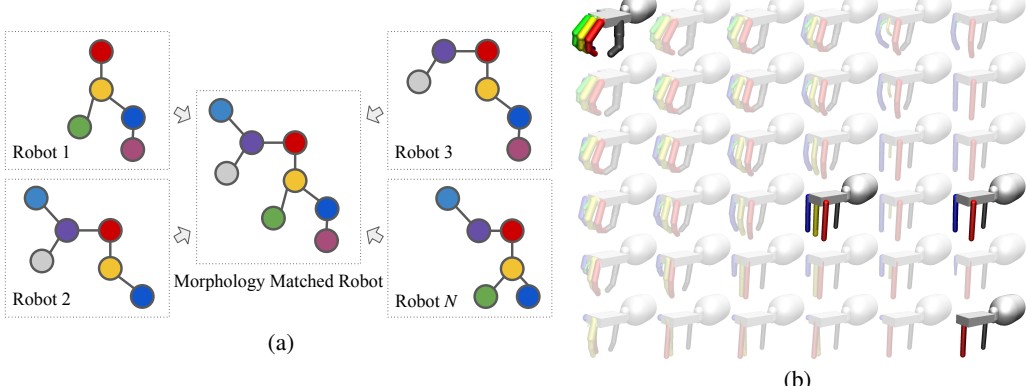

(a)

(b)

Figure 2: (a) **Morphology matching** of multiple robots. Colored circles denote corresponding robot bodies and straight lines denote robot joints. (b) An example of **robot evolution parameter space** after morphology matching of multiple robots. The four highlighted robots are the source and three target robots used in experiments in Section 5.1 respectively. Other semi-transparent robots are the generated intermediate robots.

etc. Suppose the kinematic-matched robots have $D$ physical parameters. Then each $\boldsymbol{\theta} \in \mathbb{R}^D$ uniquely defines a new robot. Suppose the physical parameters of the source robot and the $N$ target robots are $\boldsymbol{\theta}_\text{S} \in \mathbb{R}^D$ and $\boldsymbol{\theta}_{\text{T},1}, \boldsymbol{\theta}_{\text{T},2}, \ldots, \boldsymbol{\theta}_{\text{T},N} \in \mathbb{R}^D$ respectively. On each dimension, we compute the upper and lower bounds of the physical parameters

$$
\begin{aligned}
\boldsymbol{\theta}_\text{U} &= \text{MAX}(\{\boldsymbol{\theta}_\text{S}, \boldsymbol{\theta}_{\text{T},1}, \boldsymbol{\theta}_{\text{T},2}, \ldots, \boldsymbol{\theta}_{\text{T},N}\}) \\
\boldsymbol{\theta}_\text{L} &= \text{MIN}(\{\boldsymbol{\theta}_\text{S}, \boldsymbol{\theta}_{\text{T},1}, \boldsymbol{\theta}_{\text{T},2}, \ldots, \boldsymbol{\theta}_{\text{T},N}\})
\end{aligned}
\tag{2}
$$

where $\boldsymbol{\theta}_\text{U}$ and $\boldsymbol{\theta}_\text{L}$ essentially defines the *convex hull* of the set of robot physical parameters in $\mathbb{R}^D$ that encompasses the source and all target robots. We can now use continuous function $F : [0,1]^D \to \mathcal{M}$ to define an intermediate robot by interpolation between all pairs of physical parameters

$$
\boldsymbol{\theta} = (\mathbf{1} - \boldsymbol{\alpha}) \odot \boldsymbol{\theta}_\text{L} + \boldsymbol{\alpha} \odot \boldsymbol{\theta}_\text{U}
\tag{3}
$$

where $\boldsymbol{\alpha} \in [0,1]^D$ is the *evolution parameter* that describes the normalized position of a robot in the convex hull. Note that by limiting $\boldsymbol{\alpha}$ to be between $\mathbf{0}$ and $\mathbf{1}$, we assume that the convex hull is the set of all possible intermediate robots. This is a reasonable assumption since an out-of-range parameter can be physically dangerous and is also unlikely to be useful in robot continuous interpolation. The convex hull also serves as the metric space that measures the hardware difference between two robots.

In HERD (Liu et al., 2022a), the source robot and the target robot are always represented as $\mathbf{0}$ and $\mathbf{1}$ respectively because when there are only two robots, one of them must either be the lower bound or the upper bound. Different from HERD, the source and target robots in our problem are not necessarily $\mathbf{0}$ or $\mathbf{1}$. An example of the resulting robot evolution space and the positions of the source and target robots in the evolution space are illustrated in Figure 2(b).

### 3.3 ONE-TO-MANY ROBOT EVOLUTION FOR POLICY TRANSFER

Suppose the source and the $N$ target robots are represented by $F(\boldsymbol{\beta}_0) := M_\text{S}$ and $F(\boldsymbol{\beta}_1) := M_{\text{T},1}, \ldots, F(\boldsymbol{\beta}_N) := M_{\text{T},N}$ respectively where $\boldsymbol{\beta}_i \in [0,1]^D$. Similar to HERD (Liu et al., 2022a), we employ $N$ robot evolution paths $\tau_i = (F(\boldsymbol{\alpha}_{i,1}), F(\boldsymbol{\alpha}_{i,2}), \ldots, F(\boldsymbol{\alpha}_{i,K_i})), i = 1, 2, \ldots, N$ where $F(\boldsymbol{\alpha}_{i,1}) = F(\boldsymbol{\beta}_0)$ is the source robot and $F(\boldsymbol{\alpha}_{i,K_i}) = F(\boldsymbol{\beta}_i)$ is the $i$-th target robot. Following HERD, we use $K_i$ phases of policy optimizations. At phase $k$, the policy is trained on policy rollouts on robots sampled from the line $\overline{\boldsymbol{\alpha}_{i,k}\boldsymbol{\alpha}_{i,k+1}}$. The sampling window gradually converges to $\boldsymbol{\alpha}_{i,k+1}$ during training until the policy is able to achieve sufficient performance on $F(\boldsymbol{\alpha}_{i,k+1})$ before moving on to phase $k+1$. For all $k$, we set the evolution step size $||\boldsymbol{\alpha}_{i,k} - \boldsymbol{\alpha}_{i,k+1}||_p = \xi$ to be small enough so that each training phase is an easy sub-task.

Naively following HERD would require training through all the $N$ evolution paths $\tau_i$. However, if some target robots are mutually similar, at the beginning of the transfer, the robot evolution could be in roughly similar directions. Therefore, the robot evolution paths could be close to each other near the start of the paths and the policy optimization might be redundant, as illustrated in Figure 1(a).

---

**Algorithm 1** Meta-Evolve

---

**Input:** source robot $\boldsymbol{\beta}_0$ and the expert policy $\pi_{F(\boldsymbol{\beta}_0)}$ on it; target robot set $\boldsymbol{B} = \{\boldsymbol{\beta}_1, \boldsymbol{\beta}_2, \ldots, \boldsymbol{\beta}_N\}$;
**Output:** policies $\{\pi_{F(\boldsymbol{\beta}_1)}, \pi_{F(\boldsymbol{\beta}_2)}, \ldots, \pi_{F(\boldsymbol{\beta}_N)}\}$ on target robots;

---

$\{\pi_{F(\boldsymbol{\beta}_1)}, \pi_{F(\boldsymbol{\beta}_2)}, \ldots, \pi_{F(\boldsymbol{\beta}_N)}\} \leftarrow \mathsf{Meta\_Evolve}(\boldsymbol{\beta}_0, \pi_{F(\boldsymbol{\beta}_0)}, \boldsymbol{B})$
1: $\boldsymbol{\alpha} \leftarrow \boldsymbol{\beta}_0, \pi \leftarrow \pi_{F(\boldsymbol{\beta}_0)}, \Pi \leftarrow \varnothing$; // initialization
2: $\boldsymbol{\beta}_{\mathrm{Meta}}, \boldsymbol{P} \leftarrow \mathsf{Evolution\_Tree}(\boldsymbol{\alpha}, \boldsymbol{B})$; // temporary meta robot $\boldsymbol{\beta}_{\mathrm{Meta}}$; target robot partition $\boldsymbol{P} \subseteq 2^B$;
3: **while** $||\boldsymbol{\alpha} - \boldsymbol{\beta}_{\mathrm{Meta}}||_2 \geq \xi$ **do**
4:      $\pi, \boldsymbol{l} \leftarrow \arg\max_{\pi, ||\boldsymbol{l}||_2 = \xi} \quad \mathbb{E}[\rho^{\pi, F(\boldsymbol{\alpha} + \boldsymbol{l})}] - \frac{1}{2}\lambda||\boldsymbol{\beta}_{\mathrm{Meta}} - (\boldsymbol{\alpha} + \boldsymbol{l})||_p^2$; // optimize both path and reward
5:      $\boldsymbol{\alpha} \leftarrow \mathsf{MIN}(\{\mathsf{MAX}(\{\boldsymbol{\alpha} + \boldsymbol{l}, \boldsymbol{0}\}), \boldsymbol{1}\})$; // move towards meta robot, and make sure to stay within $[0, 1]$
6:      $\boldsymbol{\beta}_{\mathrm{Meta}}, \boldsymbol{P} \leftarrow \mathsf{Evolution\_Tree}(\boldsymbol{\alpha}, \boldsymbol{B})$; // update $\boldsymbol{\beta}_{\mathrm{Meta}}$ and $\boldsymbol{P}$ after robot evolution
7: **if** $|\boldsymbol{B}| = 1$ **then** // if there is only one target robot, then the meta robot is simply the target robot
8:      **return** $\{\pi\}$; // reaching the meta robot means policy transfer completes: return the policy
9: **for** $\boldsymbol{p}$ in $\boldsymbol{P}$ **do**
10:      $\Pi \leftarrow \Pi \cup \mathsf{Meta\_Evolve}(\boldsymbol{\alpha}, \pi, \boldsymbol{p})$; // recursively transfer policy in each subtree from meta robot
11: **return** $\Pi$;

---

We propose a method named Meta-Evolve that designs the $N$ evolution paths by forcing the first $m_{i,j} \in \mathbb{Z}^+$ intermediate robots to be shared between the paths towards target robots $F(\boldsymbol{\beta}_i)$ and $F(\boldsymbol{\beta}_j)$ to address the redundancy issue at the start of the training. Formally, we enforce

$$\forall k \leq m_{i,j}, \boldsymbol{\alpha}_{i,k} = \boldsymbol{\alpha}_{j,k} \tag{4}$$

This means that the two paths towards $F(\boldsymbol{\beta}_i)$ and $F(\boldsymbol{\beta}_j)$ will first reach a shared robot $F(\boldsymbol{\alpha}_{i,m_{i,j}}) = F(\boldsymbol{\alpha}_{j,m_{i,j}})$ before splitting their ways. In this way, both exploration and training overhead during policy transfer can be significantly saved due to path sharing. Theoretically, if all $N$ target robots are close enough to each other, we could expect our Meta-Evolve method to yield a speedup up to $O(N)$ compared to launching multiple one-to-one policy transfers such as HERD (Liu et al., 2022a).

As illustrated in Figure 1(b), for $N$ target robots, sharing their evolution paths essentially forms an "**evolution tree**" with its root node being the source robot and $N$ leaf nodes being the $N$ target robots. The $N - 1$ internal tree nodes are the intermediate robots that are last shared by the paths before the split. We name these internal tree nodes as "**meta robots**". Given these $N - 1$ meta robots, our Method-Evolve method first transfers the expert policy $\pi_{F(\boldsymbol{\beta}_0)}$ from the source robot $\boldsymbol{\beta}_0$ to the closest meta robot $\boldsymbol{\beta}_{\mathrm{Meta}}$ to obtain a well-trained policy, and then recursively transfer the policy towards the target robots in each sub-tree respectively. The overall idea is illustrated in Algorithm 1.

### 3.4 EVOLUTION TREE DETERMINATION

Given the source and target robots, the structure of the evolution tree and the choice of meta robots significantly impacts the overall performance of the policy transfer. However, due to the huge complexity of the robots' physical parameter and its relation to the actual MDP transition dynamics in the physical world, it is extremely difficult to develop a universal solution for the optimal evolution tree. We hereby propose the following heuristics for determining the evolution tree and meta robots.

**Evolution Tree as Steiner Tree** We aim to minimize the total $L^p$ travel distance in robot evolution parameter space from the source robot to all target robots. Mathematically, an undirected graph that interconnects a set of points and minimizes the total $L^p$ travel distance is called the $L^p$ Steiner tree or $p$-Steiner tree (Steiner, 1881; Gilbert & Pollak, 1968) of the point set. Then the evolution tree can be selected as the $p$-Steiner tree of the evolution parameter set of the source and all target robots:

$$(\boldsymbol{V}_{\mathrm{ST}}, \boldsymbol{E}_{\mathrm{ST}}) = \arg\min_{(\boldsymbol{V}, \boldsymbol{E}):\kappa((\boldsymbol{V}, \boldsymbol{E}))=1, \{\boldsymbol{\beta}_0, \boldsymbol{\beta}_1, \ldots, \boldsymbol{\beta}_N\} \subseteq \boldsymbol{V}} \sum_{(\boldsymbol{v}_1, \boldsymbol{v}_2) \in \boldsymbol{E}} ||\boldsymbol{v}_1 - \boldsymbol{v}_2||_p \tag{5}$$

where $\boldsymbol{V}_{\mathrm{ST}}$ and $\boldsymbol{E}_{\mathrm{ST}}$ are the vertex and edge sets of the $p$-Steiner tree and $\kappa(\cdot)$ denotes the graph connectivity. The neighbor(s) of the source robot acts as the initial goal(s) of the evolution. If the source robot has more than one neighbor in the tree, i.e. $\deg_{(\boldsymbol{V}_{\mathrm{ST}}, \boldsymbol{E}_{\mathrm{ST}})}(\boldsymbol{\beta}_0) > 1$, it means the evolution paths should already be split at the source robot and the policy should be transferred in each subtree respectively. The idea is illustrated in Figure 1(b) and Algorithm 2.

Note that by using $p$-Steiner tree, we assume that the training cost of transferring the policy from robot $F(\boldsymbol{\alpha}_{i,k})$ to robot $F(\boldsymbol{\alpha}_{i,k+1})$ is proportional to $||\boldsymbol{\alpha}_{i,k} - \boldsymbol{\alpha}_{i,k+1}||_p$. We believe this is a reasonable

---

**Algorithm 2** Determination of Evolution Tree and Meta Robots

---

**Input:** target robot set $\boldsymbol{B} = \{\boldsymbol{\beta}_1, \boldsymbol{\beta}_2, \ldots, \boldsymbol{\beta}_N\}$; current intermediate robot $\boldsymbol{\alpha}$;
**Output:** meta robot $\boldsymbol{\beta}_{\text{Meta}}$; target robot partition $\boldsymbol{P} \subseteq 2^{\boldsymbol{B}}$ where $\bigcup_{\boldsymbol{p} \in \boldsymbol{P}} \boldsymbol{p} = \boldsymbol{B}$ and $\forall \boldsymbol{p}_1, \boldsymbol{p}_2 \in \boldsymbol{P}, \boldsymbol{p}_1 \cap \boldsymbol{p}_2 = \varnothing$;

---

$\boldsymbol{\beta}_{\text{Meta}}, \boldsymbol{P} \leftarrow \text{Evolution\_Tree}(\boldsymbol{\alpha}, \boldsymbol{B})$
1: $(\boldsymbol{V}_{\text{ST}}, \boldsymbol{E}_{\text{ST}}) \leftarrow \arg\min_{(\boldsymbol{V}, \boldsymbol{E}): \kappa((\boldsymbol{V}, \boldsymbol{E})) = 1, \{\boldsymbol{\alpha}\} \cup \boldsymbol{B} \subseteq \boldsymbol{V}} \sum_{(\boldsymbol{v}_1, \boldsymbol{v}_2) \in \boldsymbol{E}} ||\boldsymbol{v}_1 - \boldsymbol{v}_2||_p$; // $p$-Steiner tree
2: **if** $\deg_{(\boldsymbol{V}_{\text{ST}}, \boldsymbol{E}_{\text{ST}})}(\boldsymbol{\alpha}) = 1$ **then** // the current intermediate robot has only one neighbor in the tree
3: $\quad \boldsymbol{\beta}_{\text{Meta}} \leftarrow \arg\min_{\boldsymbol{v} \in \boldsymbol{V}_{\text{ST}}} ||\boldsymbol{v} - \boldsymbol{\alpha}||_2$; // meta robot should be the neighbor
4: $\quad \boldsymbol{P} \leftarrow \{\boldsymbol{B}\}$; // there is no partition in the target robot set yet
5: **else** // the current intermediate robot has more than one neighbor, so should split paths toward each subtree
6: $\quad \boldsymbol{\beta}_{\text{Meta}} \leftarrow \boldsymbol{\alpha}$; // meta robot is the current intermediate robot itself
7: $\quad \boldsymbol{P} \leftarrow \{\boldsymbol{p}' \subseteq \boldsymbol{B} \mid \boldsymbol{p}' \subseteq \boldsymbol{V}' \subseteq \boldsymbol{V}_{\text{ST}}, \boldsymbol{E}' \subseteq \boldsymbol{E}_{\text{ST}}, \kappa((\boldsymbol{V}', \boldsymbol{E}')) = 1, \deg_{(\boldsymbol{V}', \boldsymbol{E}')}(\boldsymbol{\beta}_{\text{Meta}}) = 1\}$; // partition
8: **return** $\boldsymbol{\beta}_{\text{Meta}}, \boldsymbol{P}$;

---

assumption since the training cost should be locally proportional to the distribution difference of the MDP transition dynamics of the two robots measured in e.g. KL divergence, and should be locally proportional to the robot hardware difference $||\boldsymbol{\alpha}_{i,k} - \boldsymbol{\alpha}_{i,k+1}||_p$ when $||\boldsymbol{\alpha}_{i,k} - \boldsymbol{\alpha}_{i,k+1}||_p \to 0$, i.e. $\mathcal{D}_{KL}(F(\boldsymbol{\alpha}_{i,k}), F(\boldsymbol{\alpha}_{i,k+1})) = o(||\boldsymbol{\alpha}_{i,k} - \boldsymbol{\alpha}_{i,k+1}||_p)$.

**Implementation Details**  At each training phase of the policy transfer, the algorithm should aim to only reduce the expected *future* cost of training instead of including the past. So a more optimized implementation of our method is that, at training phase $k$, the algorithm acts *greedily* to minimize the total $L^p$ travel distance from the *current* robot $\boldsymbol{\alpha}_{i,k}$ to the meta robots $\boldsymbol{\beta}_{\text{Meta}}$ and then to all target robots $\{\boldsymbol{\beta}_i\}$ through the evolution tree. It can be implemented by replacing source robot $\boldsymbol{\beta}_0$ in Equation (5) with the current intermediate robot $\boldsymbol{\alpha}$:

$$(\boldsymbol{V}_{\text{ST}}, \boldsymbol{E}_{\text{ST}}) = \arg\min_{(\boldsymbol{V}, \boldsymbol{E}): \kappa((\boldsymbol{V}, \boldsymbol{E})) = 1, \{\boldsymbol{\alpha}\} \cup \{\boldsymbol{\beta}_1, \boldsymbol{\beta}_2, \ldots, \boldsymbol{\beta}_N\} \subseteq \boldsymbol{V}} \sum_{(\boldsymbol{v}_1, \boldsymbol{v}_2) \in \boldsymbol{E}} ||\boldsymbol{v}_1 - \boldsymbol{v}_2||_p \qquad (6)$$

This means the evolution tree and the meta robots are temporary and are updated at the start of every training phase when the robot evolution progresses, as illustrated in Algorithm 1. In practice, instead of keeping track of the entire evolution tree, we only keep the partition of the target robot set $\boldsymbol{P} \subseteq 2^{\boldsymbol{B}}$ when paths splits into subtrees and re-compute each evolution subtree after path splitting.

## 3.5 Discussions

**Can the Target Robots be Very Different?**  It is possible that target robots are in opposite directions. One extreme example is that the source robot is a five-finger hand while the two target robots are a ten-finger hand and a two-finger gripper. Our Meta-Evolve will still be able to handle such cases correctly, but the meta robot may simply be the source robot itself. This means the evolution paths are split at the start and our Meta-Evolve will be reduced to multiple runs of independent one-to-one policy transfers and does not yield any speedup in performance, which is reasonable. Fortunately, in practice, most commercial robots such as Sawyer, Panda and UR5e are indeed mutually similar in morphology and kinematics. So our Meta-Evolve can still be useful in these cases.

**Can the Meta Robots be Learned or Optimized?**  We envision the learning or optimization of the evolution tree and the meta robots being very challenging. Policy transfer through robot evolution relies on local optimization of the robot evolution. On the other hand, optimizing the evolution tree requires optimizing the robot evolution paths globally and needs an accurate "guess" of the future cost of policy transfer. In fact, our proposed heuristics can be viewed as using $L^p$ distance of evolution parameters to roughly guess the future policy transfer cost for constructing evolution tree. We leave the problem of finding the optimal evolution tree and meta robots as future work.

## 4 Related Work

**Imitation Learning across Different Robots**  Traditional imitation learning is designed for learning on the same robots (Ross et al., 2011; Ng et al., 2000; Ho & Ermon, 2016; Duan et al., 2017). However, due to a huge mismatch in transition dynamics, these works often struggle in learning across different robots. Compared to previous imitation learning methods that aim to learn across

different robots directly (Radosavovic et al., 2020; Liu et al., 2019; Rusu et al., 2015; Trabucco et al., 2022), we aim to employ robot evolution to gradually adapt the policy. Furthermore, our Meta-Evolve focuses on one-to-many imitation where the transferred policies must work on multiple target robots.

**Learning Controllers for Diverse Robot Morphology**  Recent work has studied the problem of learning a policy/controller for diverse robots. For instance, Wang et al. (2018), Huang et al. (2020) and Pathak et al. (2019) use graph neural networks to control and develop robots with different morphology that can generalize to new scenarios. Hierarchical controllers (Hejna et al., 2020) and transformers (Gupta et al., 2022; Hong et al., 2021) are also shown to be effective across diverse robot morphology. In contrast to these works, we do not co-develop the controller with morphology but transfer the policy from a source robot to multiple target robots.

**Meta-Learning**  Our Meta-Evolve is closely related to the formulation of meta-learning (Finn et al., 2017; 2018; Rajeswaran et al., 2019; Nagabandi et al., 2018; Sæmundsson et al., 2018; Schoettler et al., 2020). Different from meta reinforcement learning where only the policy $\pi$ is meta learned, our formulation can be viewed as the continuous update of both the policy $\pi$ and the transition dynamics $\mathcal{T}$ instantiated by setting different robot hardware parameters $\boldsymbol{\theta}$. Moreover, while meta-learning aims to learn a meta policy from scratch, in our problem, the source expert policy is given and used in policy transfer. Closely related to our approach is task interpolation for meta-learning (Yao et al., 2021). Different from task interpolation, our method does not require the policy to work on a range of robots at the same time but only needs each transferred policy to work on each target robot.

**Transfer Learning in RL**  Previous works on RL transfer learning have explored transferring policies by matching certain quantities across multiple tasks. Examples include learning inter-task mappings of states and actions (Gupta et al., 2017; Konidaris & Barto, 2006; Ammar et al., 2015) and cross-task reward shaping (Ng et al., 1999; Wiewiora et al., 2003). Different from these works, we do not aim to directly find the matching between different robots but gradually evolve one robot to multiple target robots through an evolution tree and transfer the expert policy along the way.

## 5 EXPERIMENTS

The design of our Meta-Evolve method is motivated by the hypothesis that by sharing the evolution paths among multiple robots through the design of the evolution tree, the overall cost of one-to-many policy transfer can be reduced compared to multiple one-to-one transfers. To show this, we apply our Meta-Evolve on the policy transfer on two types of robot learning tasks: one-to-three policy transfer on the three manipulation tasks in Hand Manipulation Suite (HMS) (Rajeswaran et al., 2018), and one-to-six policy transfer on an agile locomotion task in a maze.

### 5.1 ONE-TO-THREE MANIPULATION POLICY TRANSFER

**Source and Target Robots**  We utilize the five-finger ADROIT dexterous hand (Kumar et al., 2013) as the source robot and follow Rajeswaran et al. (2018) for the initial settings. The target robots are three robot grippers with two, three, and four fingers respectively. The target robots can be produced by gradually shrinking the fingers of the source robot, as illustrated in Figures 1 and 2(b).

**Task and RL Algorithm**  We use the three tasks from the the task suite in Rajeswaran et al. (2018): `Door`, `Hammer` and `Relocate` illustrated in Figure 3. In `Door` task, the goal is to turn the door handle and fully open the door; n `Hammer` task, the goal is to pick up the hammer and smash the nail into the board; in `Relocate` task, the goal is to pick up the ball and take it to the target position. We use a challenging sparse reward function where only the task completion is rewarded. We use NPG (Rajeswaran et al., 2017) as the RL algorithm in all compared methods. The source expert policy was trained by learning from human demonstrations collected from VR-empowered sensor glove.

**Baselines**  We compare our Meta-Evolve against three baselines: (1) *DAPG*: we launch multiple independent direct one-to-one imitation learning using DAPG (Rajeswaran et al., 2018); (2) *HERD*: we launch multiple independent one-to-one robot policy transfer with HERD; (3) *Geom-Median*: in this baseline, we allow **only one** meta robot $\boldsymbol{\beta}_{\text{Meta}}$ in the evolution tree. Mathematically, a point that minimizes the sum of $L^p$ distances to a set of points is called the $L^p$ *geometric median* of the point set. When there is only one meta robot in the Steiner tree, the meta robot $\boldsymbol{\beta}_{\text{Meta}}$ is the $L^p$ geometric median of the source and target robot evolution parameter set, i.e. $\boldsymbol{\beta}_{\text{Meta}} = \arg\min_{\boldsymbol{\beta} \in [0,1]^D} \sum_{i=0}^{N} ||\boldsymbol{\beta} - \boldsymbol{\beta}_i||_p$.

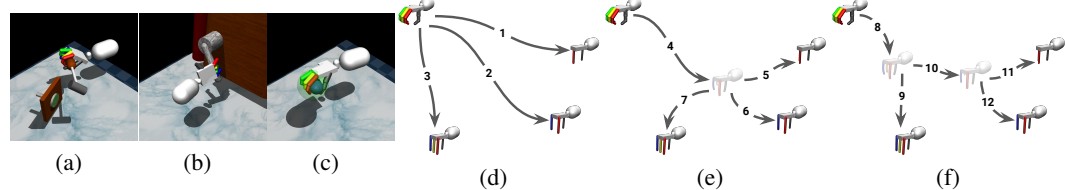

| (a) | (b) | (c) | (d) | (e) | (f) |

**Figure 3:** **Hand Manipulation Suite (HMS) tasks** (Rajeswaran et al., 2018) used in our experiments: (a) Hammer, (b) Door, and (c) Relocate. **Robot Evolution paths** of (d) using multiple independent HERD; (e) using geometric median as the only meta robot; and (f) our Meta-Evolve.

| | | | 2-finger target robot | | 3-finger target robot | | 4-finger target robot | | total | speedup |
|---|---|---|---|---|---|---|---|---|---|---|
| **Door task** | DAPG | # of train | >50K | | >50K | | >50K | | >150K | - |
| | | # of sim | >200K | | >200K | | >200K | | >600K | - |
| | | | path 1 | | path 2 | | path 3 | | total | speedup |
| | HERD | # of train | $2015 \pm 478$ | | $1016 \pm 75$ | | $1325 \pm 170$ | | $4357 \pm 395$ | $1\times$ |
| | | # of sim | $21888 \pm 2666$ | | $18612 \pm 444$ | | $23796 \pm 983$ | | $64296 \pm 1289$ | $1\times$ |
| | **Ours** | | path 8 | path 9 | path 10 | path 11 | path 12 | | total | speedup |
| | (1-Steiner | # of train | $1308 \pm 245$ | $98 \pm 17$ | $95 \pm 25$ | $527 \pm 33$ | $19 \pm 20$ | | $\mathbf{2046 \pm 200}$ | $\mathbf{2.35\times}$ |
| | Tree) | # of sim | $13964 \pm 1427$ | $964 \pm 87$ | $1784 \pm 276$ | $5628 \pm 306$ | $816 \pm 67$ | | $\mathbf{23156 \pm 1059}$ | $\mathbf{2.73\times}$ |
| | | | 2-finger target robot | | 3-finger target robot | | 4-finger target robot | | total | speedup |
| | DAPG | # of train | >50K | | >50K | | >50K | | >150K | - |
| | | # of sim | >200K | | >200K | | >200K | | >600K | - |
| | | | path 1 | | path 2 | | path 3 | | total | speedup |
| | HERD | # of train | $10323 \pm 1612$ | | $6301 \pm 1418$ | | $6513 \pm 1725$ | | $23138 \pm 4366$ | $1\times$ |
| | | # of sim | $57196 \pm 7401$ | | $41896 \pm 6629$ | | $44141 \pm 6821$ | | $143233 \pm 20362$ | $1\times$ |
| **Hammer task** | **Ours** | | path 4 | path 5 | path 6 | | path 7 | | total | speedup |
| | ($L^1$ Geom | # of train | $6295 \pm 1524$ | $5221 \pm 1972$ | $105 \pm 174$ | | $230 \pm 112$ | | $11851 \pm 3487$ | $1.95\times$ |
| | -Median) | # of sim | $32445 \pm 5697$ | $22125 \pm 7064$ | $378 \pm 592$ | | $2748 \pm 369$ | | $57696 \pm 12549$ | $2.48\times$ |
| | **Ours** | | path 8 | path 9 | path 10 | path 11 | path 12 | | total | speedup |
| | (2-Steiner | # of train | $3505 \pm 451$ | $1256 \pm 150$ | $1118 \pm 270$ | $3093 \pm 944$ | $214 \pm 110$ | | $9186 \pm 1127$ | $2.52\times$ |
| | Tree) | # of sim | $18775 \pm 1479$ | $5669 \pm 394$ | $5141 \pm 940$ | $13200 \pm 3355$ | $1646 \pm 451$ | | $44431 \pm 3908$ | $3.22\times$ |
| | **Ours** | | path 8 | path 9 | path 10 | path 11 | path 12 | | total | speedup |
| | (1-Steiner | # of train | $3848 \pm 239$ | $419 \pm 110$ | $447 \pm 191$ | $3003 \pm 1097$ | $126 \pm 132$ | | $\mathbf{7843 \pm 1380}$ | $\mathbf{2.95\times}$ |
| | Tree) | # of sim | $22603 \pm 727$ | $3007 \pm 370$ | $3566 \pm 644$ | $13421 \pm 4148$ | $1735 \pm 603$ | | $\mathbf{44333 \pm 5459}$ | $\mathbf{3.23\times}$ |
| | | | 2-finger target robot | | 3-finger target robot | | 4-finger target robot | | total | speedup |
| | DAPG | # of train | >50K | | >50K | | >50K | | >150K | - |
| | | # of sim | >200K | | >200K | | >200K | | >600K | - |
| | | | path 1 | | path 2 | | path 3 | | total | speedup |
| | HERD | # of train | $7603 \pm 1158$ | | $9657 \pm 2229$ | | $9850 \pm 932$ | | $27109 \pm 4209$ | $1\times$ |
| | | # of sim | $53568 \pm 6287$ | | $62900 \pm 6649$ | | $64244 \pm 1769$ | | $180712 \pm 14465$ | $1\times$ |
| | **Ours** | | path 4 | path 5 | path 6 | | path 7 | | total | speedup |
| **Relocate task** | ($L^1$ Geom | # of train | $9095 \pm 1616$ | $2796 \pm 715$ | $305 \pm 147$ | | $549 \pm 627$ | | $12745 \pm 2454$ | $2.13\times$ |
| | -Median) | # of sim | $39886 \pm 5641$ | $12835 \pm 2738$ | $1058 \pm 496$ | | $3972 \pm 2472$ | | $57751 \pm 9083$ | $3.13\times$ |
| | **Ours** | | path 8 | path 9 | path 10 | path 11 | path 12 | | total | speedup |
| | (2-Steiner | # of train | $5717 \pm 2097$ | $4833 \pm 244$ | $1752 \pm 285$ | $2198 \pm 18$ | $1759 \pm 111$ | | $16257 \pm 2719$ | $1.67\times$ |
| | Tree) | # of sim | $27006 \pm 7340$ | $18690 \pm 942$ | $7980 \pm 1544$ | $10242 \pm 195$ | $7452 \pm 1001$ | | $71370 \pm 11022$ | $2.53\times$ |
| | **Ours** | | path 8 | path 9 | path 10 | path 11 | path 12 | | total | speedup |
| | (1-Steiner | # of train | $9451 \pm 780$ | $467 \pm 509$ | $864 \pm 438$ | $967 \pm 206$ | $105 \pm 13$ | | $\mathbf{11853 \pm 766}$ | $\mathbf{2.29\times}$ |
| | Tree) | # of sim | $40063 \pm 3530$ | $2343 \pm 1945$ | $4869 \pm 1555$ | $7458 \pm 2940$ | $1512 \pm 141$ | | $\mathbf{56969 \pm 3750}$ | $\mathbf{3.17\times}$ |

**Table 1:** **One-to-three policy transfer experiment results on Hand Manipulation Suite tasks** (Rajeswaran et al., 2018). We use "mean ± standard deviation" from runs of five different random seeds. The details of the methods are introduced in Section 5.1. The path IDs correspond to Figures 3(d)(e)(f).

**Evaluation Metrics** For each compared method, the goal is to reach 80% success rate on all three target robots. Due to the nature of one-to-many policy transfer, the total number of RL iterations or simulation epochs it takes to reach this goal cannot be set beforehand. So we instead report the number of policy training iterations and simulation epochs needed to reach the desired success rate.

**Results and Analysis** The topology of the resulting evolution tree is illustrated in Figure 3(f). As illustrated in Table 1, in terms of the total number of training iterations needed, our Meta-Evolve method is able to achieve $2.35\times$, $2.95\times$ and $2.29\times$ improvement on the three tasks respectively compared to one-to-one policy transfer using HERD. In terms of simulation epochs needed, the improvement is $2.73\times$, $3.23\times$ and $3.17\times$ respectively on the three tasks. Direct policy transfer with DAPG never successfully completes the task, therefore the policy was never able to be trained.

Breaking down each part of the evolution paths, we observed that in our method, the paths from source robot to the meta robots are usually the most costly and constitutes the largest portion of simulation epochs and training. The cost after splitting the path at the meta robots is smaller which is the reason for smaller total cost. The baseline of using only one meta robot in the evolution tree can also yield significant improvements, however, its performance is still inferior to using an evolution tree with multiple meta robots, which shows the general tree-structured evolution paths is necessary.

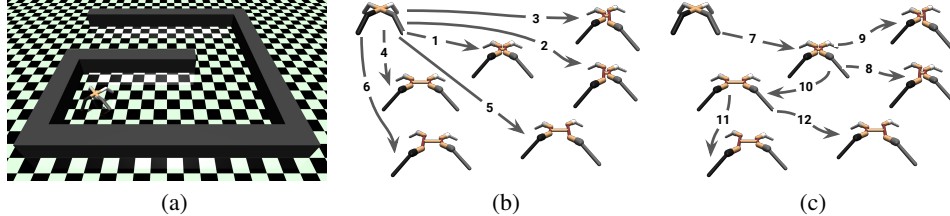

|  | (a) | (b) | (c) |

Figure 4: **Agile locomotion task in a maze**. (a) Environment and task setup; (b) Evolution paths when launching independent HERD runs; (c) Evolution paths when using $L^1$ Steiner tree as evolution tree.

|  |  | path 1 | path 2 | path 3 | path 4 | path 5 | path 6 | total | speedup |
|---|---|---|---|---|---|---|---|---|---|
| HERD | # of train | $1241 \pm 94$ | $2861 \pm 250$ | $1872 \pm 101$ | $2696 \pm 175$ | $2812 \pm 208$ | $3311 \pm 246$ | $14793 \pm 163$ | $1\times$ |
|  | # of sim | $7562 \pm 494$ | $14830 \pm 624$ | $9617 \pm 545$ | $16970 \pm 557$ | $16658 \pm 828$ | $19294 \pm 1449$ | $84931 \pm 758$ | $1\times$ |
| **Ours** |  | path 7 | path 8 | path 9 | path 10 | path 11 | path 12 | total | speedup |
| ($L^1$ Steiner | # of train | $1241 \pm 94$ | $1666 \pm 380$ | $698 \pm 156$ | $2009 \pm 352$ | $697 \pm 163$ | $428 \pm 83$ | $\mathbf{6739} \pm 721$ | $\mathbf{2.20\times}$ |
| Tree) | # of sim | $7562 \pm 494$ | $7308 \pm 1448$ | $3314 \pm 497$ | $10334 \pm 1510$ | $4051 \pm 663$ | $2354 \pm 243$ | $\mathbf{34925} \pm 2893$ | $\mathbf{2.43\times}$ |

Table 2: **One-to-six policy transfer experiment results on agile locomotion task in a maze**. We use "mean $\pm$ standard deviation" from runs of five different random seeds. The path IDs correspond to Figure 4(b)(c).

A more interesting observation is that, for some target robots and tasks, e.g. two- and three-finger target robots on Hammer task, the total cost of transferring the policy by going through multiple meta robots in the evolution tree is even smaller than the cost of directly transferring the policy to the target robot using HERD (Liu et al., 2022a). It shows that, transferring the policies to multiple related target through an evolution tree determined by our heuristic approach can possibly help each robot improve their own learning efficiency. We believe this phenomena deserve more future research attention.

**Ablation Studies** On Hammer and Relocate tasks, we provide ablation studies on the design choice of the distance measure used to construct the evolution tree, i.e. $L^1$ vs. $L^2$ distance in evolution parameter space. As illustrated in Table 1, $L^1$ distance achieves better performance than $L^2$ distance. A possible reason is that the hardware parameters mostly mutually independent, so when empirically estimating the robot transition dynamics difference, directly adding up element-wise difference may be better than using Euclidean distance which intertwines the difference on each dimension.

### 5.2 ONE-TO-SIX AGILE LOCOMOTION POLICY TRANSFER

**Experiment Settings** To show that our Meta-Evolve can generalize to diverse tasks and robot morphology, we conduct additional policy transfer experiments on an agile locomotion task illustrated in Figure 4. The goal of the robot is to move out of the maze from the starting position. The source robot is the Ant-v2 robot used in MuJoCo Gym (Brockman et al., 2016). The six target robots are four-legged agile locomotion robots with different lengths of torsos, thickness of legs, and widths of hips and shoulders. The reward function is also sparse task completion reward. We use NPG (Rajeswaran et al., 2017) as the RL algorithm. We report the number of training iterations and simulation epochs needed to reach 90% success rate on the task.

**Results and Analysis** The experiment results are illustrated in Table 2. Our Meta-Evolve method is able to achieve $2.20\times$ improvement in terms of the total training cost and $2.43\times$ total simulation cost compared to launching multiple HERD. The improvement is less compared to manipulation policy transfer. A possible reason is that locomotion tasks are less sensitive to the morphological changes of robots than manipulation tasks, therefore benefit less from our Meta-Evolve.

## 6 CONCLUSION

In this paper, we introduce a new research problem of transferring an expert policy from a source robot to multiple target robots. To solve this new problem, we introduce a new method named Meta-Evolve that utilizes continuous robot evolution to efficiently transfer the policy through an robot evolution tree defined by the interconnection of multiple meta robots and then to each target robot. We present a heuristic approach to determine the robot evolution tree. We conduct experiments on Hand Manipulation Suite tasks and an agile locomotion task and show that our Meta-Evolve can significantly outperform the one-to-one policy transfer baselines.

**Acknowledgment** Deepak Pathak is supported in part by NSF IIS-2024594 and AFOSR FA9550-23-1-0747.

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

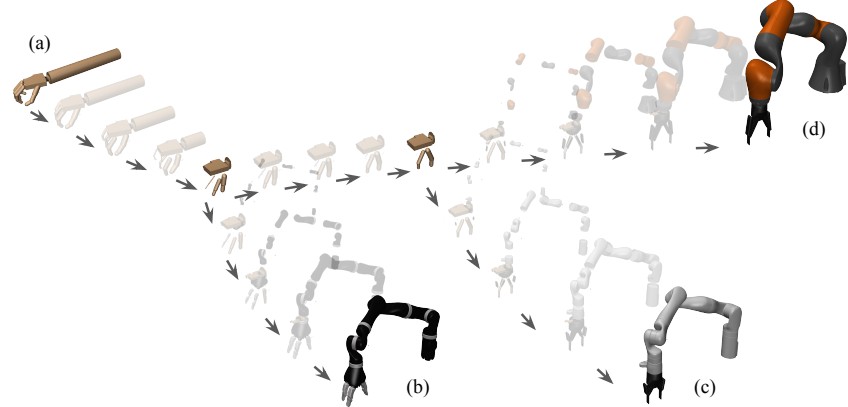

Figure 5: **The evolution tree** from the source robot, i.e. (a) **ADROIT** five-finger hand, to three target real commercial robots: (b) **Jaco** robot with three-finger Jaco gripper, (c) **Kinova3** robot with two-finger Robotiq-85 gripper, and (d) **IIWA** robot with two-finger Robotiq-140 gripper.

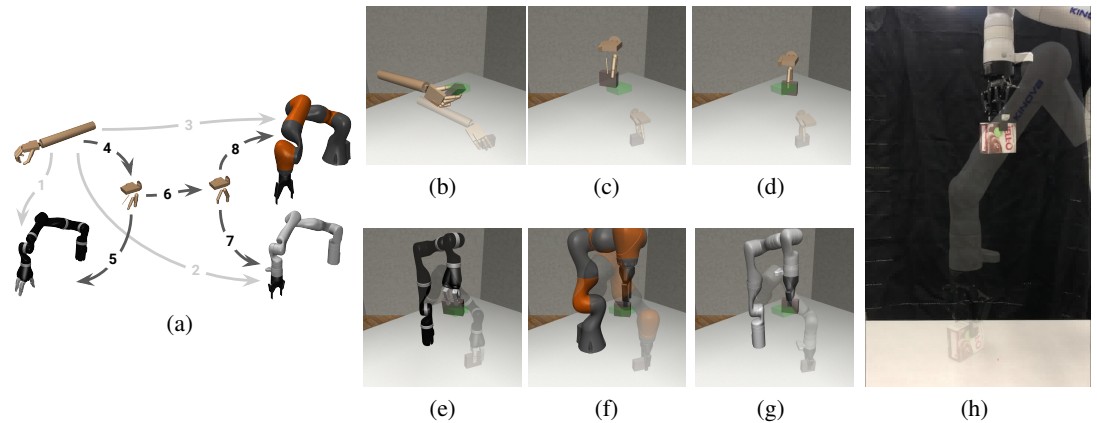

Figure 6: (a) **Robot Evolution paths** of using multiple independent HERD (paths 1, 2 and 3) and our Meta-Evolve (paths 4, 5, 6, 7 and 8); (b)-(g) **Visualization of the trained policy rollouts** on the object manipulation task. The green semi-transparent object denotes the goal position; (h) **Real-world deployment** of the target robot policy on a real Kinova3 machine.

| | | path 1 | | path 2 | | path 3 | | total | speedup |
|---|---|---|---|---|---|---|---|---|---|
| HERD | # of train | $11313 \pm 864$ | | $13405 \pm 985$ | | $13719 \pm 1077$ | | $38437 \pm 2883$ | $1\times$ |
| | # of sim | $52358 \pm 6367$ | | $65340 \pm 6693$ | | $68127 \pm 6796$ | | $185825 \pm 16731$ | $1\times$ |
| **Ours** | | path 4 | path 5 | path 6 | path 7 | path 8 | | total | speedup |
| (1-Steiner | # of train | $10810 \pm 1366$ | $4364 \pm 1609$ | $3291 \pm 1443$ | $761 \pm 220$ | $855 \pm 396$ | | $\mathbf{20081} \pm 4509$ | $\mathbf{1.91\times}$ |
| Tree) | # of sim | $48398 \pm 3811$ | $20352 \pm 2104$ | $17861 \pm 1724$ | $1569 \pm 283$ | $1744 \pm 569$ | | $\mathbf{89924} \pm 7962$ | $\mathbf{2.07\times}$ |

Table 3: **One-to-three policy transfer experiment results on DexYCB manipulation task**. We use "mean ± standard deviation" from runs of five different random seeds. The path IDs correspond to Figure 6(a).

# A    ADDITIONAL EXPERIMENTS ON REAL COMMERCIAL ROBOTS

To show that our Meta-Evolve can be applied to real robots and real-world tasks, we conduct an additional set of experiments of transferring an object manipulation policy to multiple real commercial robots.

**Source and Target Robots**  The source robot is the same ADROIT hand robot (Kumar et al., 2013) used in Section 5.1. The three target robots are as follows and are illustrated in Figure 5(b)(c)(d):

- **Jaco**: Jaco is a 7-DoF robot produced by Kinova Robotics. It is equipped with the **Jaco Three-Finger Gripper**, a three-finger gripper with multi-jointed fingers.
- **Kinova3**: Kinova3 is a 7-DoF robot produced by Kinova Robotics. It is equipped with the **Robotiq-85 Gripper**, the 85mm variation of Robotiq's multi-purpose two-finger gripper

- **IIWA**: IIWA is an industrial-grade 7-DoF robot produced by KUKA. It is equipped with the **Robotiq-140 Gripper**, the 140mm variation of Robotiq's multi-purpose two-finger gripper.

We follow the high-fidelity robot arm models introduced in Zhu et al. (2020) for the detailed physical specifications of the target robots to minimize the sim-to-real gap. Note that in the simulation model of the source ADROIT hand, the robot is rootless, i.e. connected to a virtual mount base via translation and rotation joints. We adopt the same procedure introduced in Liu et al. (2022a) to attach the ADROIT virtual mount base to the end-effector of the 7-DoF robot arm. The end-effector of the robot arm is controlled by an Operational Space Controller (OSC) (Khatib, 1987) that moves the end-effector to its desired 6D pose with PD control schema. We follow Zhu et al. (2020) for the implementation of OSC. Apart from the states of the ADROIT hand, the 6D pose of the robot end-effector is additionally included in the state of the robot. During robot evolution, the original ADROIT arm shrinks and the five-finger ADROIT hand gradually changes to be the target gripper. Please refer to Liu et al. (2022a) for more details on the idea behind this implementation. The resultant $L^1$ evolution tree is illustrated in Figure 5 and is used in our experiments.

**Task and RL Algorithms**  The task setup is illustrated in Figure 6. The goal of the robot is to pick up the object and take it to the desired goal position. The task is considered success if the distance from the object to the goal is sufficiently small. The reward function is sparse task completion reward. We use NPG (Rajeswaran et al., 2017) as the RL algorithm. The source expert policy is trained by learning from the human hand demonstrations in DexYCB dataset (Chao et al., 2021). We report the number of training iterations and simulation epochs needed to reach 80% success rate on the task.

**Results and Analysis**  The experiment results are illustrated in Table 3. Our Meta-Evolve method is able to achieve $1.91\times$ improvement in terms of the total training cost and $2.07\times$ total simulation cost compared to launching multiple HERD. This shows that our Meta-Evolve can be applied to real commercial robots. Moreover, to show that the transferred policy can be used on real target robots, we conduct real-world experiment and deploy the target robot policies on Kinova3 on the corresponding real machine as illustrated in Figure 6(h) . Please refer to the our project website for more details.

## B  ADDITIONAL DISCUSSIONS

**Can the physical parameters of all robots be known?**  It is easy to obtain all necessary physical parameters of a robot. The ultimate goal of our method is to train a policy that can be deployed on real robots. In order to do such real-world experiments, we need to obtain the robot physically. At that time, we will have access to every specification of the robot:

- **If we purchase, borrow or rent a commercial robot:** When selling their robots, licensed robot manufacturing companies would release detailed parameters of their robots. Besides, the controller software such as Operational Space Controller (OSC) (Khatib, 1987) is usually also released for the robot by the manufacturers. These controller software can only work correctly when all the necessary physical parameters of the robots are matched with the actual hardware, including the inertia and mass of every robot body, and damping and gain of every motor etc. Users with sufficient robotics expertise can easily infer the accurate physical parameters from the released control software or the manual of the robot provided to the users.
- **If we create our own robot:** Nowadays, the mechanical components of new robots are manufactured by printing 3D CAD models. Therefore, all physical parameters of the mechanical parts of the robot can be easily calculated using the 3D CAD design software. This is also how the robot manufacturing companies obtain the parameters for their own commercial robots.
- **If the robot we obtained has missing or unknown parameters:** It is not recommended to use a robot with missing or unknown parameters because it might be dangerous to do so. In the rare and extreme case where we are forced to use a robot with missing or unknown parameters, the methods for accurately measuring the parameters of unknown real robots were already developed in the 1980s (Khosla & Kanade, 1985) and have been maturally used in the robotics industry for decades since then.
- **If we attach external components to the robot:** External components may introduce additional physical parameters such as friction coefficient of the auxiliary gripper finger parts etc. These external parameters can be easily and accurately measured in lab experiments. On the other hand, it is not recommended to use an external components without knowing its detailed parameters

because it might be dangerous to do so. Please refer to related literature on mechanical or materials engineering for more details on how to measure these parameters in the lab.

**Real-robot vs. Simulation Experiments?** If we want to take advantage of the power of deep reinforcement learning in solving robotics tasks, it is imperative to collect sufficient amount of data, and performing large-scale simulation is the most convenient way to do so. On the other hand, modern simulation engines such as MuJoCo (Todorov et al., 2012), Pybullet (Coumans & Bai, 2016) and Isaac (Makoviychuk et al., 2021) allows highly accurate physics simulation to be performed. Futhermore, simulation frameworks built upon these engines such as robosuite (Zhu et al., 2020) and Orbit (Mittal et al., 2023) can simulate robot actions and robot-object interactions with very high fidelity. It is true that as long as simulation is used, there is always Sim-to-real gap. Fortunately, there has been numerous works on reducing the Sim-to-real gap in robotic control (Peng et al., 2018; Tobin et al., 2017). Sim-to-real transfer is not the focus of our work, therefore is not discussed in depth in our paper.

**Scaling/transformations on the evolution parameters $\alpha$ and its effect on Meta-Evolve?** There could be many ways to transform the physical parameter $\boldsymbol{\theta}_{\mathrm{S}}$ and $\boldsymbol{\theta}_{\mathrm{T},i}$ to be evolution parameters $\boldsymbol{\beta}_i$ by $\boldsymbol{\beta}_0 = f(\boldsymbol{\theta}_{\mathrm{S}})$ and $\boldsymbol{\beta}_i = f(\boldsymbol{\theta}_{\mathrm{T},i})$. In the main paper, we present a simple and intuitive way by normalizing each element of $\boldsymbol{\theta}$ to $[0,1]$ through a linear transformation. If $L^2$ distance is used to construct the evolution Steiner tree, the meta robots depend on the choice of the transformation function $f$. However, if $L^1$ distance is used instead, the meta robots will **not** depend on the choice of $f$, as long as $f$ is monotonic on every input dimension. As introduced in Equation (8), the $L^1$ distance of two vectors is the sum of element-wise absolute difference of the two vectors. Given the current intermediate robot $\boldsymbol{\alpha} \in \mathbb{R}^D$ and target robots $\boldsymbol{\beta}_1, \boldsymbol{\beta}_2, \ldots, \boldsymbol{\beta}_N \in \mathbb{R}^D$, the first meta robot in the Steiner tree to reach from $\boldsymbol{\alpha}$ is given by an element-wise clamp operation on $\boldsymbol{\alpha}$ to the convex hull boundaries of the target robots:

$$\boldsymbol{\beta}_{\mathrm{Meta}} = \mathrm{MAX}(\boldsymbol{\beta}_L, \mathrm{MIN}(\boldsymbol{\alpha}, \boldsymbol{\beta}_U)) \tag{7}$$

where $\boldsymbol{\beta}_U = \mathrm{MAX}(\{\boldsymbol{\beta}_1, \boldsymbol{\beta}_2, \ldots, \boldsymbol{\beta}_N\})$ and $\boldsymbol{\beta}_L = \mathrm{MIN}(\{\boldsymbol{\beta}_1, \boldsymbol{\beta}_2, \ldots, \boldsymbol{\beta}_N\})$ are the element-wise upper and lower bounds of the target robot evolution parameter convex hull. This is because the convex hull defines the spanning range of the target robot parameters and should be used as the indicator of where to split the evolution path. As long as $f$ is monotonic on every dimension of the parameter, the meta robot given by Equation (7) is invariant to $f$.

**Can Meta-Evolve generalize to any known robot configurations?** Our Meta-Evolve can generalize to any known robot configurations. For any source robot and $N$ target robots, we can always use the method described in Section 3.2 to match the robot kinematic trees and state/action spaces. Whether the policy transfer can work on that set of source and target robots depends on the task as well as the policy. For example, we do not expect a manipulation policy on a five-finger gripper can be transferred to some four-legged agile locomotion robots, though we can still define the intermediate robots between them. As long as the task and robot settings are reasonable, e.g. transferring a manipulation policy from one robot gripper to some other robot grippers, our Meta-Evolve should be able to deal with these cases, as shown by the experiments in our paper.

**Can Meta-Evolve generalize to unknown robot configurations?** Our Meta-Evolve does require knowing the configurations of the source and target robots. This is a reasonable assumption because we only deal with predetermined source and target robots. When we deploy the trained policies on the robots in the real world, we would need to first obtain these robots physically. Then at that time we will have access to everything about the robots. Please refer to the discussions on how to obtain the physical parameters of a robot, i.e. the first paragraph of Section B, for more details.

## C INTRODUCTION TO STEINER TREE

In $\mathbb{R}^D$, the Steiner tree problem is to find a network of minimum length interconnecting a set $\mathbf{B}$ of $N$ given points. Such networks can be represented by a tree. The set of the tree nodes can consist of the points in $\mathbf{B}$, known as terminals, and possibly of additional points, known as Steiner points. The length of the network is defined as the sum of the lengths of all the edges in the tree. Without allowing Steiner points, the problem is reduced to the well-known minimum spanning tree problem (Cheriton & Tarjan, 1976). Allowing additional Steiner points can possibly reduce the length of the network, but can also make the problem harder.

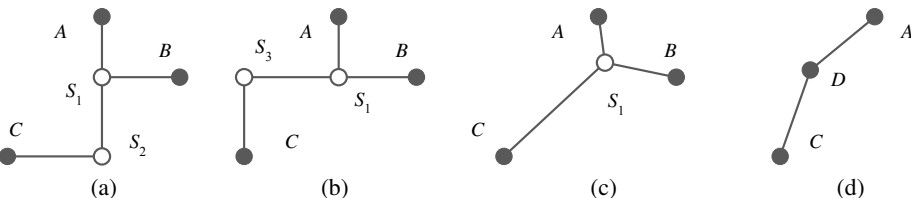

Figure 7: (a) **An $L^1$ Steiner tree** of three points $A$, $B$ and $C$ consists of additional two Steiner points $S_1$ and $S_2$; (b) **Another solution to $L^1$ Steiner tree** of the same three points $A$, $B$ and $C$ consists of two different Steiner points $S_1$ and $S_3$; (c) **The $L^2$ Steiner tree** of three points $A$, $B$ and $C$ consists of one Steiner point $S_1$ when all angles of $\triangle ABC$ are all smaller than $120°$; (d) **The $L^2$ Steiner tree** of three points $A$, $B$ and $D$ consists of no Steiner point and is reduced to an $L^2$ minimum-spanning tree when one of angles of $\triangle ABD$ is larger than $120°$.

The Steiner trees can be classified based on the metric used to measure the edge lengths. When the edge length is measured with $L^p$ norm, the Steiner tree is known as $p$-Steiner tree. In the main paper, we did ablation studies on both $L^1$ and $L^2$ norms for constructing the Steiner tree. In this section, we provide more detailed background on 1-Steiner tree and 2-Steiner tree.

### C.1  1-STEINER TREE

The $L^1$ distance of two points $\boldsymbol{\alpha} = [\alpha_1, \alpha_2, \ldots, \alpha_D] \in \mathbb{R}^D$ and $\boldsymbol{\beta} = [\beta_1, \beta_2, \ldots, \beta_D] \in \mathbb{R}^D$, also known as Manhattan distance, is the sum of element-wise absolute difference and is defined as

$$||\boldsymbol{\alpha} - \boldsymbol{\beta}||_1 = \sum_{i=1}^{D} |\alpha_i - \beta_i| \tag{8}$$

By using $L^1$ distance of evolution parameters of two robots to measure their MDP difference as the guideline for constructing Steiner tree, we not only assume the changes of each robot hardware parameter contribute equally to the change of MDP, but also assume their effect on MDP is independent and can be summed up directly. Without additional information on the actual robot hardware and task, we believe this is a reasonable assumption.

A 1-Steiner tree of a point set $\boldsymbol{B}$, also known as Rectilinear Steiner tree, is the undirected graph that interconnects $\boldsymbol{B}$ and minimizes the total $L^1$ lengths of its edges. Finding the $L^1$ Steiner tree is one of the core problems in the physical design of electronic design automation (EDA). In VLSI circuits, wire routing is only carried out by metal wires running in either vertical or horizontal directions (Sherwani, 2012).

$L^1$ Steiner tree problem is known to be an NP-hard problem. However, multiple approximate and heuristic algorithms have been introduced and used in VLSI design. Using algorithms introduced by Robins & Zelikovsky (2000), for $N$ terminal points, a good approximate solution to $L^1$ Steiner tree can be found in $O(N \log N)$ time. The solution to $L^1$ Steiner tree of a specific set of points may not be unique. Examples of $L^1$ Steiner tree are illustrated in Figure 7(a)(b).

### C.2  2-STEINER TREE

The $L^2$ distance of two points $\boldsymbol{\alpha} = [\alpha_1, \alpha_2, \ldots, \alpha_D] \in \mathbb{R}^D$ and $\boldsymbol{\beta} = [\beta_1, \beta_2, \ldots, \beta_D] \in \mathbb{R}^D$, also known as Euclidean distance, is the square root of element-wise summation of the difference squares and is defined as

$$||\boldsymbol{\alpha} - \boldsymbol{\beta}||_2 = \sqrt{\sum_{i=1}^{D} (\alpha_i - \beta_i)^2} \tag{9}$$

A 2-Steiner tree of a point set $\boldsymbol{B}$, also known as Euclidean Steiner tree, is the undirected graph that interconnects $\boldsymbol{B}$ and minimizes the total $L^2$ lengths of its edges. $L^2$ distance of evolution parameters of two robots intertwines the effect of the difference on each dimension. We believe this is one of the reasons that $L^2$ Steiner tree as the evolution tree shows worse performance in one-to-many policy transfer than $L^1$ Steiner tree.

In an $L^2$ Steiner tree, a terminal has degree between 1 and 3 and a Steiner point has degree of 3. An $L^2$ Steiner tree of $N$ terminals have at most $N - 2$ Steiner points and all Steiner points must lie in the convex hull of the terminals. The Steiner point and its three neighbors in the tree must lie in a plane, and the angles between the edges connecting the Steiner point to its neighbors are all $120°$.

$L^2$ Steiner tree problem is also known to be an NP-hard problem. Similar to $L^1$ Steiner tree, multiple approximate and heuristic algorithms have been introduced for $L^2$ Steiner tree (Fampa et al., 2016) where a good solution can be found in $O(N \log N)$ time. The Euclidean Steiner tree of three vertices of a triangle is also known as the *Fermat point* of the triangle, illustrated in Figure 7(c)(d). The solution to $L^2$ Steiner tree of a specific set of points may not be unique.

### C.3 Our Implementation

Though both $L^1$ and $L^2$ Steiner tree problems are NP-hard, fortunately, there exist multiple heuristic algorithms for approximate solutions in $O(N \log N)$ time for $N$ target robots. We used Robins & Zelikovsky (2000) for computing $L^1$ Steiner tree and Smith (1992) for computing $L^2$ Steiner tree in our implementation of finding evolution trees. In practice, we do not expect to deal with an extremely huge number of target robots. We expect the number of robots being dealt with to be under 20, which means the CPU time spent to compute both $L^1$ and $L^2$ Steiner trees is negligible.

## D Robot Evolution Specifics

For robots with different state and action spaces, Meta-Evolve converts **different state and action spaces** into **the same state and action space with different transition dynamics**. Specifically, the kinematic trees of all robots are unified by adding additional bodies and joints, though the new bodies and joints may be zero in their physical parameters, e.g. zero mass, zero sizes, zero motors etc, so that the original expert policy remains intact. The zeros are inserted to the correct positions of the state vectors of different robots to map them to the same state space. In this section, we provide more details on the evolution of the robots used in our experiments.

**Manipulation Policy Transfer Experiments** We illustrate the kinematic tree of the source ADROIT robot Kumar et al. (2013) used in our manipulation policy transfer experiments in Figure 8. During evolution, all revolute joints gradually freeze to have a motion range of 0. On the other hand, the prismatic joints are initially frozen with a motion range of 0, and some of their ranges gradually increase until the same full range.

During robot evolution, the body of the ring finger gradually shrinks to be zero-size and disappears for all target robots. Besides, other fingers may also gradually shrink and disappear for certain target robots, e.g. the middle finger and the little finger will shrink and disappear for the two-finger target robot. Our evolution solution includes the changing of $D = 65$ independent robot parameters resulting in an evolution parameter space of $[0, 1]^{65}$.

**Agile Locomotion Policy Transfer Experiments** During robot evolution of the agile locomotion policy transfer experiments, the lengths of the body, thickness of the legs, and the widths of the shoulder and hip change. This is implemented by changing the sizes of the torso frame, the size of legs as well as leg mounting positions. The solution includes the changing of $D = 5$ independent robot parameters resulting in an evolution parameter space of $[0, 1]^5$.

## E Training Details

**Hyperparameter Selection.** We present the hyperparameters of our robot evolution and policy optimization in Table 4. To fairly compare against HERD (Liu et al., 2022a), the two methods should be compared under their respective optimal performance. Fortunately, our Meta-Evolve and HERD share the same set of hyperparameters while achieving their own optimal performance. This is discovered by searching the optimal combinations of hyperparameters both methods. Another baseline method DAPG (Rajeswaran et al., 2018) uses the same RL hyperparameters illustrated in Table 4.

**Performance Threshold for Moving to the Next Intermediate Robot.** We used success rate as the indicator for deciding whether to move on to the next intermediate robot during policy transfer.

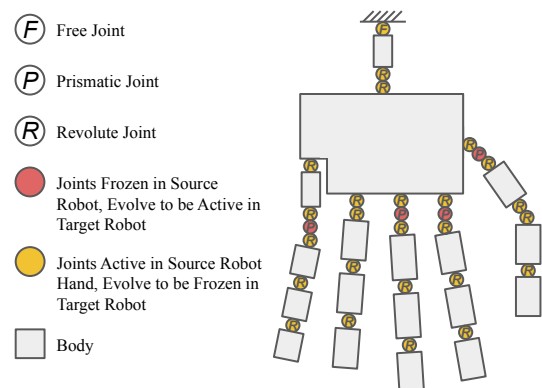

Figure 8: **Kinematic tree of dexterous hand robot**. All revolute and free joints will gradually freeze during evolution. The two prismatic joints are initially frozen and evolve to be active.

| Hyperparameter | Value |
|---|---|
| RL Discount Factor $\gamma$ | 0.995 |
| GAE | 0.97 |
| NPG Step Size | 0.0001 |
| Policy Network Hidden Layer Sizes | (32,32) |
| Value Network Hidden Layer Sizes | (32,32) |
| Simulation Epoch Length | 200 |
| RL Traning Batch Size | 12 |
| Evolution Progression Step Size $\xi$ | 0.03 |
| Number of Sampled Evolution Parameter Vectors for Jacobian Estimation in HERD Runs | 72 |
| Evolution Direction Weighting Factor $\lambda$ | 1.0 |
| Sample Range Shrink Ratio | 0.995 |
| Success Rate Threshold for Moving to the Next Training Phase | 66.7% |

Table 4: **The value of hyperparameters** used in our experiments.

Specifically, the policy transfer moves on to the next intermediate robot if and only if the success rate on the current intermediate robot exceeded a certain threshold. As illustrated in Table 4, 66.7% is roughly the best option for both HERD and Meta-Evolve. Using a higher success rate threshold may waste training overhead on intermediate robots and slow down policy transfer, while using a lower success rate threshold may sacrifice sample efficiency in later stages of policy transfer due to sparse-reward settings.

**Evaluation Metrics.** We followed HERD (Liu et al., 2022a) and adopted *the training overhead needed to reach 80% success rate* as our evaluation metric. The core idea of using this evaluation metric is to compare the efficiency of the policy transfer when the difficulty of transferring each policy is unknown beforehand. Using an alternative success rate higher than 80% as the evaluation metric could also be feasible. However, since the success rate threshold for moving on to the next intermediate robot is 66.7%, the success rate increase from 80% to a higher one can only happen **after** the policy transfer is completed. Therefore, the remaining part of training for reaching a higher success rate is simply vanilla reinforcement learning on the target robots and is irrelevant to our problem of inter-robot policy **transfer**, so should not be included in the evaluation.

**Training Platforms.** We use PyTorch (Paszke et al., 2019) as our deep learning framework and NPG (Rajeswaran et al., 2017) as the RL algorithm in all manipulation policy transfer and agile locomotion transfer experiments. We used MuJoCo (Todorov et al., 2012) as the physics simulation engine.

## F  VISUALIZATIONS

On the three Hand Manipulation Suite (Rajeswaran et al., 2018) tasks, we provide visualizations for the expert policy on the source robot and the transferred policies on three target robots in Figure 9. As shown in the visualizations, during policy transfers, the original behaviors can be generally

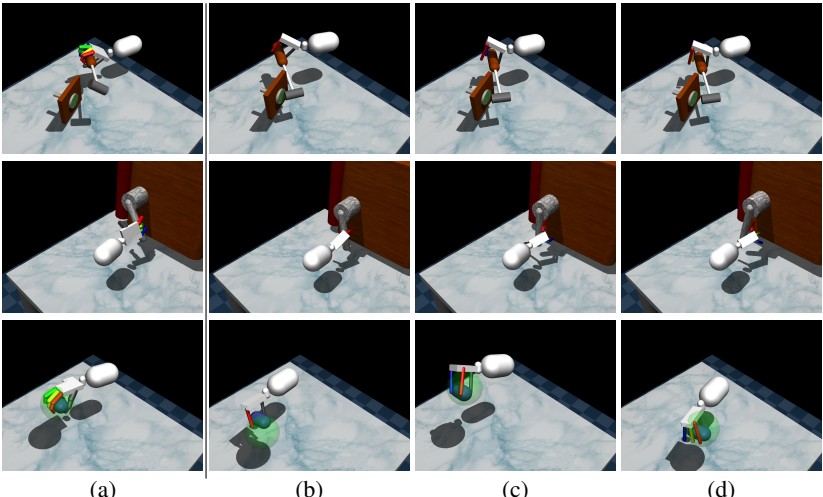

|      |      |      |      |
| :--: | :--: | :--: | :--: |
| (a)  | (b)  | (c)  | (d)  |

Figure 9: **Visualization of the trained policy rollouts on Hand Manipulation tasks Suite** (Rajeswaran et al., 2018). From the first to the third row: `Hammer` task, `Door` task and `Relocate` task. From left to right: (a) source robot; (b) 2-finger target robot; (c) 3-finger target robot; (d) 4-finger target robot.

maintained in the original expert policy and also transfer it to each target robot. Please refer to the attached supplementary video or our project website for more details on the visualizations.

