# OpenReview forum: "Meta-Evolve: Continuous Robot Evolution for One-to-many Policy Transfer"
_ICLR.cc/2024/Conference — ICLR 2024 poster_

### Official Review · Reviewer_Escv · 2023-11-01

**Soundness:** 3 good
**Presentation:** 3 good
**Contribution:** 3 good
**Rating:** 5
**Confidence:** 4

**Summary:**

This paper introduces a method to perform expert policy transfer from one robot to many other robots of different morphology. As opposed to previous works, this method defines a tree structure that finds a meta-robot to act as an intermediary fine-tuning platform before the policy is transferred to a target robot (i.e. finding a common robot-morphology ancestor in the robot evolution tree). This robot evolution tree allows for faster transfer between source and target robots because the policy only needs to be transferred to the meta robot once, as opposed to N times (as with previous methods). The paper experiments on several transfer tasks, including manipulation tasks where the morphology of the robotic hand changes from source to target.

**Strengths:**

This paper is clearly written and motivates the problem well. This method exposes clear gaps in the literature regarding and proposes a simple and novel method that can improve the sample efficiency of existing robot-robot transfer methods. Additionally, the authors propose an interpretable way to measure the distance between robot hardware morphologies in equation (3).

Mainly, the authors' proposal of an ancestral meta-robot common to all target robots is an interesting idea worth exploring. It is clear from the results reported in the paper that this idea speeds up training time/simulation time.

**Weaknesses:**

There are several weakness in the paper that I think need to be addressed.

1.
-  In section 3.2, the authors discuss kinematic tree matching between robots with similar kinematic structures. However, I am concerned that they are claiming that just because two robots share a similar kinematic tree, that they also share a similar expert policy. In my understanding, the kinematic tree matching would relate a pair of robots with 2 fingers and 3 fingers more closely than a pair robot with 2 fingers and 4 fingers. However, it very well may be the case that a robot with 2 fingers will act more similarly to a robot with 4 fingers than one with 3 fingers.
- Further, the convex hull of robot hardwares might be irrelevant if the robots close in hardware-space are far in policy space

2. I am curious about the method the authors use to actually train the policies on the new robots. The description of generating new robots is clear but the method for actually transferring the policies is not.

3. I am concerned about the notion of neural plasticity in this problem. Simply fine-tuning the policy more times might lead to the degredation in performance that we see in the HERD and REvolveR methods

4. The authors do not demonstrate the quality of the new policies. I know they are training to achieve some specified success rate but I think more experimentation on what the transferred policies are is important to discuss in this paper

**Questions:**

1. How does changing the morphplogy of a robot affect the optimal policy?
2. How are the authors training the new policies on the new robots?
3. Does the simple fact that the authors are fine-tuning the same policy more times on the baselines lead to poor performance?
4. Can you please describe quantitatively/qualitatively the quality of the learned target policies between the different methods?

---

> ### Author Response · Authors · 2023-11-15
> **Thank you for your time and review! (1/2)**
>
> We thank Reviewer Escv for their insightful comments and the time they spent on reviewing our paper. Regarding the questions and comments by Reviewer Escv:
>
> ---
>
> > `However, I am concerned that they are claiming that just because two robots share a similar kinematic tree, that they also share a similar expert policy.`
>
> We apologize for the confusion. To clarify, we didn’t claim the relationship between similar kinematic trees and similar expert policies.
>
> Our assumption is that, if two robots are similar in their *morphology and dynamics* (specified by their kinematic structures, physical parameters etc.), they may also be similar in their optimal policy and have small cost of policy transfer between them. It’s not just about the kinematic trees. We will include additional sentences in the next version of the paper to make it more clear.
>
> ---
>
> > `In my understanding, the kinematic tree matching would relate a pair of robots with 2 fingers and 3 fingers more closely than a pair robot with 2 fingers and 4 fingers. However, it very well may be the case that a robot with 2 fingers will act more similarly to a robot with 4 fingers than one with 3 fingers.`; `Further, the convex hull of robot hardwares might be irrelevant if the robots close in hardware-space are far in policy space`.
>
> We agree that it is possible that in some cases, transferring policy to robots closer in hardware space can actually be more expensive.
>
> Our assumption is that the policy transfer cost is **locally** proportional to robot hardware space distance when the hardware difference is small. Globally, this assumption may not be true.
>
> Based on this assumption, we propose a heuristics to use physical parameter distance to estimate the cost of local policy transfer. The heuristics are used to construct evolution trees and are shown to be effective in our extensive experiments. As mentioned in the 2nd paragraph of Section 3.5, we leave the problem of finding the optimal way of estimating policy transfer cost and constructing evolution trees as future work:
>
> “*Policy transfer through robot evolution relies on local optimization of the robot evolution. On the other hand, optimizing the evolution tree requires optimizing the robot evolution paths globally and needs an accurate “guess” of the future cost of policy transfer. In fact, our proposed heuristics can be viewed as using $L^p$ distance of evolution parameters to roughly guess the future policy transfer cost for constructing evolution tree. We leave the problem of finding the optimal evolution tree and meta robots as future work.*”
>
> ---
>
> > `I am curious about the method the authors use to actually train the policies on the new robots. The description of generating new robots is clear but the method for actually transferring the policies is not.`; `How are the authors training the new policies on the new robots?`
>
> Sorry for the confusion. As mentioned in the 1st paragraph of Section 3.3 and line 4 of Algorithm 1, the policy is transferred by fine-tuning on the next intermediate robots, i.e. training the policy on the next intermediate robots using an RL algorithm. In our experiments, we use NPG algorithm as our RL algorithm.
>
> ---
>
> > `I am concerned about the notion of neural plasticity in this problem. Simply fine-tuning the policy more times might lead to the degradation in performance that we see in the HERD and REvolveR methods.`; `Does the simple fact that the authors are fine-tuning the same policy more times on the baselines lead to poor performance?`
>
> We did a sufficient amount of experiments on REvolveR, HERD and our Meta-Evolve, and we did not see performance degradation problems.
>
> ---
>
> > `The authors do not demonstrate the quality of the new policies. I know they are training to achieve some specified success rate but I think more experimentation on what the transferred policies are is important to discuss in this paper`; `Can you please describe quantitatively/qualitatively the quality of the learned target policies between the different methods?`
>
> Our method aims at improving the overall policy transfer efficiency. After policy transfer is completed, the performance/quality of transferred policies can be further improved by continuing to train on the target robots with RL algorithms.
>
> Given sparse task-completion reward, success rate is the best way to evaluate the policy quality. Quantitatively, transferred target policies reach the same success rate at the end of policy transfer, though it takes much smaller overall training/simulation cost for our Meta-Evolve to do so than HERD.
>
> Qualitatively, as suggested, we visualize the transferred policies on Hammer and Relocate tasks using different methods in this video: https://drive.google.com/file/d/1Ln8oLZhU3g1hw_lZhqAu-CC0rW2WvSW_/view?usp=sharing (select 1080p for best quality). As shown in the video, policies transferred using different methods show slightly different behaviors, though they are all able to complete the tasks.

---

> > ### Author Response · Authors · 2023-11-15
> > **Thank you for your time and review! (2/2)**
> >
> > > `How does changing the morphology of a robot affect the optimal policy?`
> >
> > Changing the morphology of a robot can affect the optimal policy in different ways, depending on factors including how the robot morphology is changed and the task definition.
> >
> > We assume that, in most cases, the amount of change of optimal policy is **locally** proportional to the amount of change of robot morphology.

---

> > > ### Author Response · Authors · 2023-11-20
> > >
> > > Dear Reviewer Escv,
> > >
> > > Thank you for the comments and suggestions! As the discussion period is ending, we would appreciate it if you could kindly check our response. Please do not hesitate to contact us if there are other clarifications we can offer. Thanks!

---

### Official Review · Reviewer_e334 · 2023-11-01

**Soundness:** 3 good
**Presentation:** 3 good
**Contribution:** 2 fair
**Rating:** 6
**Confidence:** 4

**Summary:**

The paper discusses the Meta-Evolve method, which aims to transfer expert policies from a source robot to multiple target robots. The paper introduces a new research problem and proposes a solution that utilizes continuous robot evolution and a robot evolution tree. The experiments conducted on hand manipulation tasks and agile locomotion tasks show that Meta-Evolve outperforms one-to-one policy transfer baselines. Ablation studies and discussions on handling different target robots and learning or optimizing the evolution tree are also presented. Overall, the paper highlights the effectiveness of Meta-Evolve in inter-robot policy transfer and suggests areas for future research.

**Strengths:**

The paper highlights the effectiveness of Meta-Evolve in inter-robot policy transfer, as demonstrated through experiments on hand manipulation tasks and agile locomotion tasks (very interesting). The results show that Meta-Evolve outperforms one-to-one policy transfer baselines in terms of training and simulation costs.

**Weaknesses:**

The experiments are mainly limited to hand manipulation tasks, which can be easily represented by tree structures. Does your main idea still works on modular robots (like 3d voxel-based robot)?

Reference: Nick Cheney, Robert MacCurdy, Jeff Clune, and Hod Lipson. Unshackling evolution: evolving soft robots with multiple materials and a powerful generative encoding. In GECCO ’13, 2013.

**Questions:**

Can and how your method transfer to the real world robotic problems? Like manipulation tasks.

---

> ### Author Response · Authors · 2023-11-13
> **Thank you for your time and review!**
>
> We thank Reviewer e334 for their insightful comments and the time they spent on reviewing our paper. Regarding the questions and comments by Reviewer e334:
>
> ---
>
> > `The experiments are mainly limited to hand manipulation tasks, which can be easily represented by tree structures. Does your main idea still works on modular robots (like 3d voxel-based robot)?`
>
> The main assumption of our method is that the robot can be defined by several continuous parameters so that the robots can be continuously interpolated by varying the parameters and the robot evolution tree can be constructed in high-dimensional space $[0,1]^D$.
>
> For modular and soft robots, as long as the robot can be defined by multiple continuous parameters, our main idea should still be able to work.
>
> ---
>
> > `Can and how your method transfer to the real world robotic problems? Like manipulation tasks.`
>
> Yes, it can.
>
> We showed that our method can be applied to real-world robots. In Section A of the appendix (page 13), we included experiments where we transfer a manipulation policy from a five-finger dexterous robot hand to three real commercial robots: Jaco, Kinova3 and IIWA. Please refer to Section A for more details.
>
> The videos of the experiments in Section A and the **real-robot demo** can be found on our anonymous project website: https://sites.google.com/view/meta-evolve.

---

> > ### Comment · Reviewer_e334 · 2023-11-19
> > **Thank you for your response**
> >
> > Thanks for your response, I have no further questions.

---

> > > ### Author Response · Authors · 2023-11-19
> > > **Thank you!**
> > >
> > > Thank you so much for your comment. Just wondering if there is anything else that we can do to further address your concerns so that you are able to raise your score?

---

### Official Review · Reviewer_NXFw · 2023-11-01

**Soundness:** 4 excellent
**Presentation:** 4 excellent
**Contribution:** 3 good
**Rating:** 8
**Confidence:** 3

**Summary:**

In Meta-Evolve: Continuous Robot Evolution for One-to-many Policy Transfer, the authors introduce a novel method of policy transfer which they claim is able to improve the efficiency of policy transfer between robots when multiple recipient bodies are involved, by performing policy transfer along an “evolutionary” line of intermediary bodies. Using “kinematic tree matching” and some geometric heuristics (based on Steiner trees), they are able to identify bodies which can be used in the paths of multiple recipients, reducing the total number of bodies where training is needed to enable the policy transfer. They demonstrate significant increases in efficiency over alternative methods which do not use such intermediates.

**Strengths:**

The paper is very well written with nice graphical explanations of the technique and clear and well reasoned theoretical work. The authors were careful to make sure that the technical details were written unambiguously and the descriptions of the underlying mathematics were excellent. The results of the work seem compelling and are well described.

**Weaknesses:**

It is not obvious to me, as someone who does not deal much with the realm of physical robots, how common the problem addressed in this paper is—my uninformed guess would be that it is not so common, but the technique does not suffer much for this. It is not obvious that the work has much to do with evolution, in either the biological or computational senses, except in its use of trees which are reminiscent of the tree of life. It is also not obvious why the word “meta” was chosen, especially given that it has other connotations in Reinforcement Learning. It is not established within the paper how efficacious the kinematic tree matching methods are—although it seems to be good enough at transferring, it would be interesting to see how this matching deals with adverse cases. The number of training runs in the experiments was quite small, with only 5 random seeds. It would have been better to see a higher-quality evaluation of the method, but the presented results seem clear. It is also not obvious why some of the thresholds in the evaluation were selected, and it would be interesting to see if a change from 80% to 70% or 90% would change the order of the methods. I suspect that it would not, but the choice of selecting a single threshold leaves the question open.

**Questions:**

1. How common is the problem addressed by this paper, of transferring an effective policy from one robot morphology to multiple others at once?
2. Why was the only the 1-Steiner tree attempted in the door task?
3. Why were so few seeds used for the results?
4. Why was the value of 80% selected?
5. Did the authors consider the relationship between the reduction in the total length of the paths in the graph (as compared to independent paths) and the reduction in the training time? Were these correlated?
6. Why wasn’t DAPG run to completion? How effectively was it able to perform the task, given that it did not complete in the time provided?
7. Do the authors have any comments on why (e.g. in the Hammer task, on path 8-10-11) their outperformed HERD on one-to-one transfer?

---

> ### Author Response · Authors · 2023-11-16
> **Thank you for your time and review!**
>
> We thank Reviewer NXFw for their insightful comments and the time they spent on reviewing our paper. Regarding the questions and comments by Reviewer NXFw:
>
> ---
>
> > `1. How common is the problem addressed by this paper, of transferring an effective policy from one robot morphology to multiple others at once?`
>
> The problem is common and will become even more popular in the future.
>
> One example: recent advances in the robot learning community started to research on learning across multiple different robot embodiments, such as the recently released Open X-Embodiment benchmark [A] (https://robotics-transformer-x.github.io/) which includes **22** different robots.
>
> [A] Padalkar, Abhishek, et al. "Open x-embodiment: Robotic learning datasets and rt-x models.". arXiv preprint arXiv:2310.08864 (2023). 2023 Oct 13.
>
> ---
>
> > `2. Why was only the 1-Steiner tree attempted in the door task?`
>
> This is because after ablation studies on Hammer and Relocate tasks, it already shows that 1-Steiner tree is the best design choice of our Meta-Evolve.
>
> ---
>
> > `3. Why were so few seeds used for the results?`; `The number of training runs in the experiments was quite small, with only 5 random seeds.`
>
> We conducted statistical analysis and found that with 5 seeds, our method Meta-Evolve is better than the baseline HERD with more than 99.99% confidence, on all tasks.
>
> This means 5 random seeds are enough for our experiments.
>
> ---
>
> > `4. Why was the value of 80% selected?`; `it would be interesting to see if a change from 80% to 70% or 90% would change the order of the methods`
>
> We explained this in the 3rd paragraph of Section E of the appendix:
>
> “*We followed HERD (Liu et al., 2022a) and adopted the training overhead needed to reach 80% success rate as our evaluation metric. The core idea of using this evaluation metric is to compare the efficiency of the policy transfer when the difficulty of transferring each policy is unknown beforehand. Using an alternative success rate higher than 80% as the evaluation metric could also be feasible. However, since the success rate threshold for moving on to the next intermediate robot is 66.7%, the success rate increase from 80% to a higher one can only happen after the policy transfer is completed. Therefore, the remaining part of training for reaching a higher success rate is simply vanilla reinforcement learning on the target robots and is irrelevant to our problem of inter-robot policy transfer, so should not be included in the evaluation.*”
>
> ---
>
> > `5. Did the authors consider the relationship between the reduction in the total length of the paths in the graph (as compared to independent paths) and the reduction in the training time? Were these correlated?`
>
> Yes, the two reductions are positively correlated.
>
> ---
>
> > `6. Why wasn’t DAPG run to completion? How effectively was it able to perform the task, given that it did not complete in the time provided?`
>
> During exploration on new robots, DAPG has trouble finding successful trajectories with non-zero reward due to the huge robot difference. This means extremely low sample efficiency under sparse task-completion reward setting, and means that DAPG is not effective in directly transferring the expert policies to new robots.
>
> ---
>
> > `7. Do the authors have any comments on why (e.g. in the Hammer task, on path 8-10-11) their outperformed HERD on one-to-one transfer?`
>
> Sorry we don’t have a good explanation for that. We leave the detailed study of this phenomenon as a future work.
>
> ---
>
> > `It is not obvious that the work has much to do with evolution, in either the biological or computational senses`; `It is also not obvious why the word “meta” was chosen, especially given that it has other connotations in Reinforcement Learning`.
>
> The word "evolution" has more than just biological or computational meaning. According to Webster's Dictionary (the most authoritative dictionary for modern English), another meaning of "evolution" is "a process of change in a certain direction" (see the second listed meaning in this link: https://www.merriam-webster.com/dictionary/evolution).
>
> The current name for our method “Meta-Evolve” is a combination of “meta” (inspired by the idea of meta learning where the learning objective is across multiple tasks) and “evolve” (to show the connection to robot evolution).
>
> We are open to changing the name to a better one if Reviewer NXFw could suggest a better name for our method.

---

> > ### Comment · Reviewer_NXFw · 2023-11-20
> > **Response**
> >
> > 1. Thank you.
> > 2. Thank you - I would appreciate seeing the other designs attempted in all of the tasks; it is strange to exclude one of them from only one task.
> > 3. I suspect that the assumptions necessary to come to such a conclusion after only 5 trials are poorly warranted and would very much like to see the sample size increased unless a compelling explanation of the choice of statistical test can be provided.
> > 4. I would still be interested in a comparison of multiple values.
> > 5. It would be good to see a graph of this data in a final version of the paper.
> > 6. It's disappointing, but not your fault, that the other method failed to deliver a reasonable comparison.
> > 7. I would really like to see this, as I think it would be extremely interesting and could (if it finds something interesting) shed significant light on the reason that Meta-Evolve led to such significant improvement; it is possible (though perhaps unlikely) that Meta-Evolve achieved superiority over the comparator methods because of a factor other than the kinetic trees.
> >
> > Lastly, on the evolution point: I am generally skeptical of the use of biological terms in computer science, and I think that the word evolution, in a usage such as this, is not really separable from the biological connotations. I would prefer something like "Skill Transfer Trees: Continuous Training Regimens for One-to-many Morphology Policy Transfer", but this is by no means crucial to my rating of the paper.

---

> > > ### Author Response · Authors · 2023-11-22
> > > **Thank you for your additional comments! (1/2)**
> > >
> > > We thank Reviewer NXFw for the additional comments. Regarding the additional comments by Reviewer NXFw:
> > >
> > > ---
> > >
> > > > `2. Thank you - I would appreciate seeing the other designs attempted in all of the tasks; it is strange to exclude one of them from only one task.`
> > >
> > > We agreed it is better to run the ablation study for all three experiments. We will add the ablation studies on the Door task, and will include it in the final version of the paper. Thank you for the suggestion!
> > >
> > > We want to kindly point out that the existing results already showed our proposed method with 1-Steiner tree clearly outperforms baselines. The goal of running additional ablation studies is to re-confirm that 1-Steiner tree design is the best one among other variance of our approach, which has been shown in the Hammer and Relocate tasks.
> > >
> > > ---
> > >
> > > > `3. I suspect that the assumptions necessary to come to such a conclusion after only 5 trials are poorly warranted and would very much like to see the sample size increased unless a compelling explanation of the choice of statistical test can be provided.`
> > >
> > > Thanks for the follow-up question. Our assumption is that the performance (i.e. training and simulation cost) of both our method (1-Steiner) and baseline HERD are both Gaussian distributions and are mutually independent. Based on this assumption, we ran two-sample T-tests for the null hypothesis that the mean of our Meta-Evolve’s performance is larger than the mean of HERD’s performance.
> > >
> > > Table 1. p-values of the two-sample T-tests:
> > >
> > > | Door | Hammer | Relocate | Maze |
> > > | --- | --- | --- | --- |
> > > $1.51 \times 10^{-5}$ | $9.54 \times 10^{-6}$ | $3.08 \times 10^{-6}$ | $8.63 \times 10^{-9}$
> > >
> > > As shown in Table 1, the p-values of the tests are smaller than 0.0001 on all four tasks, which means with more than 99.99% confidence, our method is better than baseline HERD. This is significant as we run hypothesis tests on all four tasks and have consistent high confidence.
> > >
> > > Table 2. p-values of the two-sample T-tests after scaling our method (1-Steiner) by $1.5\times$:
> > >
> > > | Door | Hammer | Relocate | Maze |
> > > | --- | --- | --- | --- |
> > > $7.41 \times 10^{-4}$ | $1.58 \times 10^{-4}$ | $2.42 \times 10^{-3}$ | $1.17 \times 10^{-5}$
> > >
> > > We also tried to scale the performance numbers of our method by $1.5\times$, and we found that on all tasks, the p-values are still smaller than 0.01 on all tasks, as shown in Table 2. This means with more than 99% confidence, our method is better than baseline HERD by at least $1.5\times$.
> > >
> > > To check if the assumption of Gaussian/normal distribution stands, on the Hammer task, we ran 15 additional random seeds to make it 20 random seeds in total. We conducted a Shapiro-Wilk Test on the 20 samples and got a p-value of 0.689. This means there is not enough evidence to reject our normality assumption. So we believe it is OK to justify the Gaussian assumption.
> > >
> > > We hope the above statistical tests are sufficient to address your concerns on the number of random seeds.
> > >
> > > ---
> > >
> > > > `4. I would still be interested in a comparison of multiple values.`
> > >
> > > As suggested, we provide the performance of baseline HERD and our Meta-Evolve (1-Steiner) with 70% terminal success rate threshold on the Hammer and Relocate tasks as follows. Note that the results reported in the paper are obtained using an 80% terminal success rate threshold.
> > >
> > > Table 3. *Hammer* task under different terminal success rate thresholds:
> > >
> > > | success rate | evaluation metrics | HERD | Ours (1-Steiner) | speedup |
> > > | --- | --- | --- | --- | --- |
> > > 80% | \# of train | 23138 ± 4366 | 7843 ± 1380 |  2.95x
> > > 80% | \# of sim | 143233 ± 20362 | 44333 ± 5459 | 3.23x
> > > 70% | \# of train | 23001 ± 4328 | 7813 ± 1363 | 2.94x
> > > 70% | \# of sim | 141593 ± 20031 | 44232 ± 5397 | 3.20x
> > >
> > > Table 4. *Relocate* task under different terminal success rate thresholds:
> > >
> > > | success rate | evaluation metrics | HERD | Ours (1-Steiner) | speedup |
> > > | --- | --- | --- | --- | --- |
> > > 80% | \# of train | 27109 ± 4209 | 11853 ± 766 |  2.29x
> > > 80% | \# of sim | 180712 ± 14465 | 56969 ± 3750 | 3.17x
> > > 70% | \# of train | 26872 ± 3905 | 11561 ± 429 | 2.32x
> > > 70% | \# of sim | 179529 ± 13193 | 55989 ± 2521 | 3.21x
> > >
> > > As illustrated in the above two tables, the difference of results between using 70% and 80% terminal success rates is almost negligible.
> > >
> > > The reason for such a small difference is that given the 70% success rate already being sufficiently high, it is easy for the success rate to increase to an even higher value (e.g. 80%) on the target robots due to sufficiently high sample efficiency.

---

> > > > ### Author Response · Authors · 2023-11-22
> > > > **Thank you for your additional comments! (2/2)**
> > > >
> > > > > `5. It would be good to see a graph of this data in a final version of the paper.`
> > > >
> > > > This is a good idea! Here is the plot of this data on the Hammer task: https://drive.google.com/file/d/1HgoMuHkSzZg9RAZ9_fN_-q4IHWXpNVN4/view?usp=sharing. In the plot, the X axis is the total path length and the Y axis is the total number of training iterations. The error bars correspond to the standard deviations from runs of different random seeds. This figure shows that there is a positive correlation.
> > > >
> > > > As you suggested, we will include this graph in the final version of the paper. Thank you for your suggestion.
> > > >
> > > > ---
> > > >
> > > > > `7. I would really like to see this, as I think it would be extremely interesting and could (if it finds something interesting) shed significant light on the reason that Meta-Evolve led to such significant improvement; it is possible (though perhaps unlikely) that Meta-Evolve achieved superiority over the comparator methods because of a factor other than the kinetic trees.`
> > > >
> > > > We don’t have a good explanation.
> > > >
> > > > Our best guess is that on the Hammer task, our Meta-Evolve serves as a good regulator so that the optimizer was able to find a path with a smoother loss landscape during policy optimization than the one-to-one baseline HERD. The landscape analysis of high dimensional space itself is a wide open area as shown in the following literature [B] for example. While it might be challenging to fully verify it since the loss landscape is related to the task, policy as well as the intermediate robot, we agree it is an interesting research direction for future work.
> > > >
> > > > [B] Engstrom, Logan, et al. "Exploring the landscape of spatial robustness." International conference on machine learning. PMLR, 2019.
> > > >
> > > > ---
> > > >
> > > > > `I would prefer something like "Skill Transfer Trees: Continuous Training Regimens for One-to-many Morphology Policy Transfer"`
> > > >
> > > > Excellent suggestion! We agree this is a much clearer title.
> > > >
> > > > ---
> > > >
> > > > Thank you again for spending time on our paper. Please let us know if there is anything else that we can do to further address your concerns.

---

### Official Review · Reviewer_6cdy · 2023-11-08

**Soundness:** 3 good
**Presentation:** 3 good
**Contribution:** 2 fair
**Rating:** 5
**Confidence:** 5

**Summary:**

The paper presents a novel method for transfer learning of policies from one source robot to multiple target robots. In other to achieve this, the authors extend HERD (and REvolveR) with a novel robot kinematic evolution tree that is based on Steiner Trees. The experiments showcase that the proposed method performs better than HERD and other baselines in several experiments.

**Strengths:**

- The idea is using p-Steiner trees to represent the robot kinematic evolution tree is interesting, intuitive, simple and seems effective.
- The method clearly accelerates transfer learning compared to the baselines provided.
- The paper is generally well-written and the main messages are effectively conveyed.

**Weaknesses:**

- My main concern is the practicality of the proposed method. In other words, how can this be used in real(ish) world applications? I explain further:
    - First, as the name suggests the method explores in the kinematic space of the robots. What happens if the dynamics differ drastically? An example would be, having a robot that has the same kinematic structure but twice the masses. Can the method handle this? It seems that no. Isn't changing the dynamics but keeping the kinematics the same a different robot? I think this deserves more intuition and explanation.
    - Then, I am not sure how we should interpret the additional experiments on commercial robots. I cannot see the added value of those experiments and how they contribute towards convincing us that the method is applicable to real-world situations/robots. The robots are purely position-controlled and the behaviors are quite simple.
    - Lastly, the comments on Sim2Real are weak imho. I do not see why the paper is not relevant to Sim2Real methods. The paper/method claims one to many policy transfer with *different dynamics* involved (per robot).
- The authors have chosen to consider a model-free approach to RL/transfer learning, while obviously the models are known (at least the kinematics part). This choice should be better motivated and highlighted in the text.
- **One of the videos is not properly anonymized; we can clearly see the face of someone performing the experiments.** Is this part of the video part of the DexYCB dataset? If this is the case, it should have been highlighted. If not (and the one performing the experiments is one of the authors), the AC should take a position on this as I am not sure if this is allowed by ICLR regulations.

**Questions:**

1) How can the proposed method handle significant dynamics differences between the source and target robots? Can it even do that?
2) How does the proposed method fit inside the Sim2Real literature?
3) How can the proposed method integrate model-based RL/learning? Why did the authors not experiment with this?

---

> ### Author Response · Authors · 2023-11-16
> **Thank you for your time and review!**
>
> We thank Reviewer 6cdy for their comments and the time they spent on reviewing our paper. Regarding the questions and comments by Reviewer 6cdy:
>
> ---
>
> > `as the name suggests the method explores in the kinematic space of the robots. What happens if the dynamics differ drastically? An example would be, having a robot that has the same kinematic structure but twice the masses. Can the method handle this? It seems that no. Isn't changing the dynamics but keeping the kinematics the same a different robot?`
> > `How can the proposed method handle significant dynamics differences between the source and target robots? Can it even do that?`
>
> Sorry for the confusion. To clarify, our method also explores different robot dynamics because it also explores different robot physical parameters such as mass and inertia, as described in Section 3.2 and Equation (3).
>
> Our method can handle the case where the source and target robot dynamics differ, but of course the difference cannot be unreasonably large. For example, we cannot expect a manipulation policy on a multi-fingered robot gripper to be transferred to a multi-legged locomotion robot dog.
>
> As long as the dynamics difference is reasonable, our method is able to handle that. This is because our method interpolates physical parameters after kinematic tree matching, which is essentially interpolating robot dynamics.
>
> As mentioned by Reviewer 6cdy, we conduct an additional experiment where we double the mass of the agile locomotion robots. Both HERD and our method can still successfully transfer the policy to the new target robots with double masses.
>
> ---
>
> > `I am not sure how we should interpret the additional experiments on commercial robots. I cannot see the added value of those experiments and how they contribute towards convincing us that the method is applicable to real-world situations/robots.`
>
> The goal of our method is to transfer an expert policy from one robot to multiple other predefined robots.
>
> In the additional experiments in Section A of the appendix, we verified that our method can transfer a manipulation policy from a five-finger dexterous robot hand to three different existing commercial robots (Jaco, Kinova3, IIWA) in simulation while outperforming baseline HERD by roughly 2$\times$ in terms of overall efficiency. In addition, we verified that the transferred policy on Kinova3 in simulation can be deployed on the real Kinova3 machine in a real-world situation.
>
> Therefore, the additional experiments support our claim and further show that our method can be applied to real-world situations/robots.
>
> ---
>
> > `the comments on Sim2Real are weak imho. I do not see why the paper is not relevant to Sim2Real methods.`
> > `How does the proposed method fit inside the Sim2Real literature?`
>
> Conceptually, sim-to-real transfer is not within the scope of our paper:
>   - Our Meta-Evolve deals with transferring policy from the *source robot* to *target robots*, in *simulation*.
>   - Sim-to-real transfer deals with transferring the policies on *target robots* from *simulation* to *real machine*.
>
> Therefore, it is actually OK to not even discuss sim-to-real transfer in our paper.
>
> We still discussed it anyway in Section B (and also conducted additional real-robot experiments in Section A as experimental evidence), simply to address people’s (potential) concerns over deploying the transferred policies to real machines.
>
> ---
>
> > `The authors have chosen to consider a model-free approach to RL/transfer learning, while obviously the models are known (at least the kinematics part). This choice should be better motivated and highlighted in the text.`
> > `How can the proposed method integrate model-based RL/learning? Why did the authors not experiment with this?`;
>
> In general, models are not easily known in robotic tasks even if the robot kinematics and dynamics are known. This is because there are unknown robot-object interactions that affect both the robot states and the object states. Robot-object interactions are not only determined by the robots but also depend on the objects in the environment.
>
> It’s easy to apply model-based RL algorithms in our method though. We can just instantiate the policy optimization part of line 4 of Algorithm 1 with a model-based RL algorithm.
>
> However, we believe our existing experiments and ablation studies are already sufficient to support the claims made in our paper.
>
> ---
>
> > `One of the videos is not properly anonymized; we can clearly see the face of someone performing the experiments. Is this part of the video part of the DexYCB dataset? If this is the case, it should have been highlighted.`
>
> **This part of the video is from public dataset DexYCB** (other example videos of the dataset can be found in https://dex-ycb.github.io/).
>
> Also, this information has already been highlighted in the supplementary video. At the start of the video (0:00:00-0:00:03), we have already highlighted with text “**DexYCB Dataset Human Demonstration**”.

---

> > ### Author Response · Authors · 2023-11-20
> >
> > Dear Reviewer 6cdy,
> >
> > Thank you for the comments and suggestions! As the discussion period is ending, we would appreciate it if you could kindly check our response. Please do not hesitate to contact us if there are other clarifications we can offer. Thanks!

---

> > ### Comment · Reviewer_6cdy · 2023-11-22
> >
> > > To clarify, our method also explores different robot dynamics because it also explores different robot physical parameters such as mass and inertia
> >
> > First, there was no confusion. My comment actually says "The method explores the kinematic parameters and not the dynamic parameters". In other words, there is no systematic way that the difference in dynamics is explored.
> >
> > > Therefore, the additional experiments support our claim and further show that our method can be applied to real-world situations/robots.
> >
> > Having more experiments is always "helpful" and nice, but I still do not really see the contribution here. Too many important details are missing. Is this position control? Torque control? Velocity? What's the exact state? Overall, too many details are missing to evaluate the experiments.
> >
> > > Our Meta-Evolve deals with transferring policy from the source robot to target robots, in simulation. Sim-to-real transfer deals with transferring the policies on target robots from simulation to real machine.
> >
> > I believe that proposed method is very close to the Sim2Real literature. Sim2Real is an "umbrella" term covering the transferring of policies from a source environment to a target environment. The target environment is usually the real world but this is not necessary. Your Meta-Evolve deals with transferring the policy from a source robot to a target robot. These read and look like very similar.
> >
> > As you mention you choose to test it in simulation. Why? We need a much stronger motivation about the practicality of the method.
> >
> > > simply to address people’s (potential) concerns over deploying the transferred policies to real machines.
> >
> > I already expressed my concerns regarding the contribution of these experiments. There are very little details. And from the video, this looks like position control, which has very close to zero reality gap.
> >
> > > This part of the video is from public dataset DexYCB
> >
> > Thanks for the clarification.

---

> ### Author Response · Authors · 2023-11-22
> **Thank you for your additional comments! (1/2)**
>
> We thank Reviewer 6cdy for the additional comments. Regarding the additional comments by Reviewer 6cdy:
>
> ---
>
> > `My comment actually says "The method explores the kinematic parameters and not the dynamic parameters". In other words, there is no systematic way that the difference in dynamics is explored.`
>
> We kindly point out that the definition of “kinematic parameters” and “dynamic parameters” are: “*The Denavit-Hartenberg parameters constitute the kinematic parameters, and the link masses, link inertias, and center-of-mass vectors are the dynamic parameters.*” (1st paragraph of Section 2 of [A])
>
> Our method explores robot physical parameters including mass and inertia, as described in Section 3.2 and Equation (3). Therefore our method also explores different robot dynamics.
>
> [A] Pradeep Khosla and Takeo Kanade. "Parameter Identification of Robot Dynamics." CDC 1985.
>
> ---
>
> > `Having more experiments is always "helpful" and nice, but I still do not really see the contribution here.`;
> > `I already expressed my concerns regarding the contribution of these experiments. … this looks like position control, which has very close to zero reality gap.`
>
> The focus of our paper is to transfer policies between robots with different morphologies, which we found in our experiments are already non-trivial even with zero sim-to-real gap. Sim-to-real is also an important and actively-studied field but not our focus nor contribution of this paper. The goal of the real-robot demo includes:
>  - To verify our assumption that our method tolerates a relatively small level of uncertainties (e.g. model errors, delay and noise from controller) required in real world deployment.
>  - For the demonstration purpose for the broader robotic community.
>
> We also thank Reviewer 6cdy for acknowledging that the robot and the controller “`has very close to zero reality gap`”. This can help justify the value of our additional experiments: to verify that the transferred policies in simulation can be deployed on the real machines.
>
> ---
>
> > `Too many important details are missing. Is this position control? Torque control? Velocity? What's the exact state? Overall, too many details are missing to evaluate the experiments.`;
> > `There are very little details. And from the video, this looks like position control...`
>
>   - Control: we use the  Operational Space Controller (OSC) [B] that moves the end-effector to desired 6D pose with PD control schema. It is also detailed in the robotsuite package: https://robosuite.ai/docs/modules/controllers.html#Operational-Space-Control---Pose-with-Fixed-Impedance.
>   - Robot state: states of the ADROIT hand + 6D pose of the robot end-effector
>
> We follow the default robosuite setting for the robot arm, therefore, we did not specify the details but refer to the readers to the documentations of robosuite. We have updated our paper to include the above details in the 3rd paragraph of Section A. We are happy to provide more details and will release source code to ensure the work is reproducible.
>
> [B] O. Khatib. "A unified approach for motion and force control of robot manipulators: The operational space formulation." IEEE Journal on Robotics and Automation, 1987.
>
> ---
>
> > `Sim2Real is an "umbrella" term covering the transferring of policies from a source environment to a target environment. The target environment is usually the real world but this is not necessary. Your Meta-Evolve deals with transferring the policy from a source robot to a target robot. These read and look like very similar.`
>
> We follow the definition of “Sim2Real” in literature such as [C,D,E] where the term “Sim2Real” specifically refers to transferring from simulation to its counterpart real systems. While in our setting, the source and target robots are different even with zero reality gap which some people refer to as “domain transfer”. We agree from a very high level, they are all transferring between different distributions of uncertainties.
>
> In the context of this paper specifically, we believe there are two steps of transfer learning: (1) from one robot A to another robot B; (2) from simulation of B to physical deployment of B. These two steps are different as the uncertainties have different sources. In (1), the uncertainties come from the different morphologies of robots, while in (2) the uncertainties come from some uncontrollable factors that are hard to precisely incorporate in the simulation e.g. the delay of motors. We would be happy to review the sim-to-real domain and clarify the difference from and similarities with our approach.
>
> [C] J. Truong et al. "Rethinking sim2real: Lower fidelity simulation leads to higher sim2real transfer in navigation." CoRL 2023.
>
> [D] M. Kaspar et al.. "Sim2real transfer for reinforcement learning without dynamics randomization." IROS 2020.
>
> [E] P. M. Scheikl et al. "Sim-to-real transfer for visual reinforcement learning of deformable object manipulation for robot-assisted surgery." RA-L 2022.

---

> > ### Author Response · Authors · 2023-11-22
> > **Thank you for your additional comments! (2/2)**
> >
> > > `As you mention you choose to test it in simulation. Why?`
> >
> > Quoted from the previous answer “*In the context of this paper specifically, we believe there are two steps of transfer learning: (1) from one robot A to another robot B; (2) from simulation of B to physical deployment of B.*” The benefit of using simulation is that we could decouple the uncertainties in these two steps, therefore better understand the strengths and limitations of the proposed method.
> >
> > ---
> >
> > > `We need a much stronger motivation about the practicality of the method.`
> >
> > Thanks for pushing us to think more about it. Some examples of the use case of our approach:
> >   - When owning multiple types/brands of robots, engineers can use our approach to efficiently train multiple different robots to (collaboratively) do the same/similar tasks.
> >   - When hardware engineers design multiple new layouts of a robotic hardware, they can use our approach to quickly generate a control policy and evaluate the functionality with much better data efficiency.
> >   - A robot in a harsh/resource limited environment (e.g. a space robot on a different planet) may repair itself using parts (e.g. fingers) of another robot. Our method may increase the survival rate of the robots in a harsh/resource limited environment and make the robot more resilient to change to its hardware configuration.
> >
> > We would be happy to modify our motivation part to make the motivation clearer.

---

> > > ### Comment · Reviewer_6cdy · 2023-11-22
> > >
> > > Thank you for the additional comments/discussion!
> > >
> > > > “The Denavit-Hartenberg parameters constitute the kinematic parameters, and the link masses, link inertias, and center-of-mass vectors are the dynamic parameters.” (2nd paragraph of Section 2 of [A])
> > >
> > > I cannot find this sentence in the uploaded pdf (I interpreted [A] as appendix A, but I searched the whole document for the sentence). I re-downloaded, but still wasn't able to find it.
> > >
> > > > Our method explores robot physical parameters including mass and inertia, as described in Section 3.2 and Equation (3).
> > >
> > > I honestly did not get this and I have read the paper many times. Maybe the authors should make this more explicit? Or include experiments that further showcase this?
> > >
> > > > We also thank Reviewer 6cdy for acknowledging that the robot and the controller “has very close to zero reality gap”. This can help justify the value of our additional experiments: to verify that the transferred policies in simulation can be deployed on the real machines.
> > >
> > > I did not mean it like this. I meant that showcasing that a position-controlled policy transfers to the real robot is not interesting for manipulators. This is well known and does not showcase anything for the new method at hand.
> > >
> > > > The focus of our paper is to transfer policies between robots with different morphologies, which we found in our experiments are already non-trivial even with zero sim-to-real gap.
> > >
> > > I understand this, but what I am saying is that Sim2Real methodology can be used (and has already been used) for transferring policies between different morphologies/domains. As I described in my comment, Sim2Real methods are broader than the proposed method/target problem; Sim2Real methods basically include the specific instantiation of the problem in this paper. Thus it makes sense to include comparisons. The comparisons would also validate (or not) the need for a novel specific method (aka in case generic Sim2Real methods are not enough, which could definitely be the case).
> > >
> > > > We follow the definition of “Sim2Real” in literature such as [C,D,E] where the term “Sim2Real” specifically refers to transferring from simulation to its counterpart real systems.
> > > > We agree from a very high level, they are all transferring between different distributions of uncertainties.
> > >
> > > I think I made myself clear. We can define infinite many smaller sub-problems (and name them differently), but not all of them are interesting/meaningful to solve. For me domain adaptation/Sim2Real/policy transfer/[put another nice name here that refers to the same problem] are solving the same (high-level if you wish) problem. Then, each paper/novel method that attempts to solve a sub-problem should motivate their choice with theoretical or empirical evidence. I believe (with the current state of the manuscript) that the authors should motivate their specific sub-problem with empirical evaluation as well.
> > >
> > > > The benefit of using simulation is that we could decouple the uncertainties in these two steps, therefore better understand the strengths and limitations of the proposed method.
> > >
> > > Yes including sim to sim experiments is interesting and valuable. Why the authors chose to not compare to generic Sim2Real methods is not clear to me.
> > >
> > > > We would be happy to modify our motivation part to make the motivation clearer.
> > >
> > > Yes, re-writing the introduction/motivation can help the reader appreciate the work more and position it better in the literature. My concerns about comparing to more generic Sim2Real methods stand though even if with better motivation.

---

> > > > ### Comment · Reviewer_6cdy · 2023-11-22
> > > >
> > > > > I cannot find this sentence in the uploaded pdf (I interpreted [A] as appendix A, but I searched the whole document for the sentence). I re-downloaded, but still wasn't able to find it.
> > > >
> > > > I just realized that [A] meant the reference given! Sorry for this. In that case, similar text should be present in the main text of the paper and made clear in both Section 3.2 and the experiments section/details.

---

> ### Author Response · Authors · 2023-11-23
> **Thanks for additional discussions!**
>
> Thanks for the additional discussions. Regarding Reviewer 6cdy’s concerns over comparison to generic Sim2Real methods/methodology:
>
> Sim2Real transfer can be viewed as a special case of domain transfer. It specifically deals with the problem of transferring knowledge/skills from simulation to its counterpart real-world systems. Some common assumptions regarding the domain difference in Sim2Real:
>   1. **Unknown Domain Difference**: One assumption is that the precise nature and extent of the domain difference are not explicitly known. This implies that there are discrepancies between the simulation and the real world, but the exact details of these differences are not identified or quantified.
>   2. **Small Range of Domain Difference**: Another assumption is that the domain difference is within a relatively small range. This assumption suggests that, despite the differences, the simulation is considered precise enough for the purposes of training, and the variations between the simulated and real-world environments are manageable.
>   3. **Parameter Errors as the Cause**: Sim2Real often assumes that the primary cause of domain difference lies in errors in the model parameters or limitations in the simulation's fidelity. In other words, the assumption is that if the simulator were perfect and the parameters were precisely tuned, the domain difference would be minimal.
>
> These assumptions justifies the most popular Sim2Real method *domain randomization*, which randomizes the domain parameters by adding a small random perturbation noise in order to train a domain-robust policy that can generalize to a range of domains.
>
> However, in our problem, while the 3rd assumption of Sim2Real still holds true, the 1st and 2nd assumption does not. In our problem:
>   1. **Known Domain Difference**: The domain differences between the source and target robots are assumed to be known due to the source and target robots being pre-defined.
>   2. **Huge Domain Difference**: The domain differences between the source and target robots can be extremely huge due to huge differences in morphology.
>
> Our approach is specifically designed for the assumption of our problem:
>   1. We construct robot evolution trees based on known robot physical parameters (known domains). This allows policy transfer to be along paths with known domain directions which is very likely to be better than completely random perturbations of domains.
>   2. We decompose the task of transferring the policy to an extremely different domain into multiple sub-tasks by introducing multiple intermediate robots (intermediate domains). This is especially beneficial for policy transfer in challenging sparse task-completion reward settings where maintaining sufficient performance is crucial for sample efficiency and the overall policy transfer efficiency.
>
> Therefore, we believe that, in our problem, even if a Sim2Real method can work, it is very likely to show much worse overall performance than our method.
>
> We hope the above discussions are sufficient to address your concerns on the comparison of Sim2Real and our method.

---

### Meta-Review · Area_Chair_4Mar · 2023-12-13

**Metareview:**

This is an interesting paper describing a novel method that is confusingly named. The name and terminology makes the reader think it is about evolutionary robotics, i.e. the use of evolutionary algorithms to develop robot morphology and/or policy. Instead, this paper is about policy transfer (using gradient-based methods) between multiple robots on the same "evolutionary tree"; as far as I can tell, no actual evolutionary algorithm is used. It seems to me that this misunderstanding underlies several other misunderstandings from at least one of the reviewers. (It is also possible that I have misunderstood the paper.)

The main contribution is a way of transferring policies between physically different robots that are "evolutionarily related". This is done by transferring many times to interpolations between the robots in question. The interpolations happen on a graph, and a "meta-robot" is defined to connect the various robots on the graph efficiently.

One weakness of the paper is the writing; while the English is good and the reasoning clear, concepts are not necessarily defined where they should be. The reader is left to fill in their own interpretation of overloaded terms such as "evolution" and "model".

It seems to me that the paper is strong enough for acceptance.

**Justification For Why Not Higher Score:**

The problem addressed is not of great generality.

**Justification For Why Not Lower Score:**

The method works and is novel.

---

### Decision · Program_Chairs · 2024-01-16

Accept (poster)